# FedCL: Critical Learning Periods-aware Adaptive Client Selection in Federated Learning

## Abstract

Federated learning (FL) is a distributed optimization paradigm that learns from data samples distributed across a number of clients. Adaptive client selection that is cognizant of the training progress of clients has become a major trend to improve FL efficiency but not yet well-understood. Most existing FL methods such as FedAvg and its state-of-the-art variants implicitly assume that all learning phases during the FL training process are equally important. Unfortunately, this assumption has been revealed to be invalid due to recent findings on critical learning (CL) periods, in which small gradient errors may lead to an irrecoverable deficiency on final test accuracy. In this paper, we develop FedCL, a CL periods-aware FL framework to reveal that adaptively augmenting exiting FL methods with CL periods, the resultant performance is significantly improved when the client selection is guided by the discovered CL periods. Experiments based on various machine learning models and datasets validate that the proposed FedCL framework consistently achieves an improved model accuracy while maintains comparable or even better communication efficiency as compared to state-of-the-art methods, demonstrating a promising and easily adopted method for tackling the heterogeneity of FL training.

## 1 Introduction

Federated learning (FL) (McMahan et al., 2017) has emerged as an attractive distributed learning paradigm that leverages a large number of clients to collaboratively learn a joint model with decentralized training data under the coordination of a centralized server. In contrast with centralized learning, the FL architecture allows for preserving clients' privacy and reducing the communication burden caused by transmitting data to the server. While there is a rich literature in distributed optimization in the context of machine learning, FL distinguishes itself from traditional distributed optimization in two key challenges: high degrees of system and statistical heterogeneity (Kairouz et al., 2019).

In an attempt to address the heterogeneity and improve the efficiency of FL, various optimization methods have been developed for FL. In particular, the federated averaging algorithm (FedAvg) (McMahan et al., 2017) is the current state-of-the-art method for FL. In each communication round, FedAvg leverages local computation at each client and employs a centralized server to aggregate and update the global model parameter. While FedAvg has demonstrated empirical success in heterogeneous settings, it fails to fully address the underlying challenges associated with heterogeneity. For example, FedAvg randomly selects a subset of clients in each iteration regardless of their statistical heterogeneity, which has been shown to diverge empirically in settings where data samples of each client follow a non-identical and independent distribution (non-IID).

A recent trend of improving FL efficiency focuses on adaptive client selection during the FL training process, such as (Ruan et al., 2021; Karimireddy et al., 2020; Li et al., 2020a; Wang et al., 2020c;b; Cho et al., 2020; Wang et al., 2020a; Rothchild et al., 2020; Lai et al., 2021). However, these studies implicitly assume that all learning phases during the FL training process are equally important. Unfortunately, this assumption has recently been revealed to be invalid due to the existence of *critical learning (CL) periods*, i.e., the final quality of a deep neural network (DNN) model is determined by the first few training epochs, in which deficits such as low quality or quantity of training data will cause irreversible model degradation. Notably, this phenomenon was revealed in the latest series of works (Achille et al., 2019; Jastrzebski et al., 2019; Golatkar et al., 2019; Jastrzebski et al., 2021) for centralized learning, and in (Yan et al., 2022) for FL settings. Despite their insightful findings, there

remains to be a major *gap* between the observation of CL periods in FL and the goal of *more efficient training* and *improved model accuracy*, since existing client selection methods in state-of-the-art FL algorithms are *unaware* of the existence of CL periods in FL, which were only identified using a computationally expensive metric that emerges after the full training process.

In this paper, we close this gap by demonstrating the importance of *CL periods awareness* for client selection in state-of-the-art FL algorithms. Through a range of carefully designed experiments on different machine learning models and datasets, we observe a consistently improved model accuracy without sacrificing communication efficiency by augmenting state-of-the-art FL algorithms with CL periods. We build upon recent work by (Yan et al., 2022), who showed that if the training dataset for each client is not recovered to the entire training dataset early enough in the training process, the test accuracy of FL is permanently impaired. We extend this notation to client selection in FL and show that a larger number of clients are only required during these CL periods. As a result, an adaptive and efficient client selection scheme is akin to finding CL periods in the FL training process. These CL periods can be detected in an online manner using a new metric called Federated Gradient Norm (FGN). To the best of our knowledge, this is the first step taken towards exploiting CL periods for adaptive client selection in FL to mitigate heterogeneity.

Our main contributions in this paper are summarized as follows:

1. We propose a practical, easy-to-compute Federated Gradient Norm (FGN) metric to identify CL periods in an online manner, fixing a major paradox for connecting CL periods with client selection for the efficient FL training goal.

2. We propose a simple but powerful CL periods-aware FL framework, dubbed as FedCL, that is generic across and orthogonal to different FL methods. In particular, we use FedAvg as our building block since it is the first and the most widely used one. FedCL inspects the changes in FGN to detect CL periods in FL training process, and adaptively determines the number of clients to participate in each FL training round. With extensive empirical evaluation on different machine learning models with different datasets, we show that FedCL consistently achieves up to 11% accuracy improvement while maintaining comparable or even better communication efficiency compared to FedAvg.

3. We show that the CL periods awareness can be easily combined with state-of-the-art FL methods, such as FedProx (Li et al., 2020a), VRL-SGD (Liang et al., 2019) and FedNova (Wang et al., 2020c). When augmented by FedCL via manipulating the client selection, existing methods achieve up to 11%, 13% and 10% accuracy improvement, respectively, compared to training without the awareness of CL periods.

## 2 RELATED WORK

**Critical Learning (CL) Periods.** The presence of CL periods in centralized neural network training was first highlighted in (Achille et al., 2019; Jastrzebski et al., 2019). Some other works (Golatkar et al., 2019; Jastrzebski et al., 2021; Frankle et al., 2020; Jastrzebski et al., 2020) have also highlighted the importance of early training phase in centralized learning. The existence of CL periods in FL was recently discovered in (Yan et al., 2022). However, studying CL phenomena hinged on costly information metric (e.g., eigenvalues of the Hessian) that emerges after the full training, limiting their practical benefits. We differ from existing works by developing an easy-to-compute metric to identify CL periods during the training process in an online manner.

**Federated Learning and Client Selection.** The state-of-the-art method for FL is FedAvg, which was first proposed in (McMahan et al., 2017) and has sparked many follow-ups (Stich, 2018; Wang & Joshi, 2021; Yu et al., 2019) with full client participation. In practice, only a small fraction of clients participate in each training round, which exacerbates the effect of data heterogeneity. As a result, solutions with partial client participation and effect of data heterogeneity have been developed and analyzed (Katharopoulos & Fleuret, 2018; Ruan et al., 2021; Karimireddy et al., 2020; Li et al., 2020a; Wang et al., 2020c; Li et al., 2020b; Wang et al., 2020b; Ribero & Vikalo, 2020; Cho et al., 2020; Wang et al., 2020a; Yang et al., 2021; Cho et al., 2022; Reddi et al., 2021; Haddadpour & Mahdavi, 2019; Khaled et al., 2020; Stich & Karimireddy, 2020; Woodworth et al., 2020; Horváth & Richtarik, 2021; Nishio & Yonetani, 2019; Malinovskiy et al., 2020; Pathak & Wainwright, 2020; Goetz et al., 2019; Tang et al., 2022). For a comprehensive introduction to FL and other algorithmic variants

---

**Algorithm 1** FedAvg

---

**Input:** $\mathcal{M}, \eta, E, \boldsymbol{\theta}^{(0)}, T$

1: **for** $t = 0, 1, \cdots, T - 1$ **do**
2:     Server selects a subset $\mathcal{M}^{(t)}$ of $\mathcal{M}$ clients at random
3:     Server sends $\boldsymbol{\theta}^{(t)}$ to all selected clients
4:     Client $k \in \mathcal{M}^{(t)}$ updates $\boldsymbol{\theta}^{(t)}$ via $E$ iterations of SGD on $\mathcal{D}_k$ with stepsize $\eta$ to obtain $\boldsymbol{\theta}_k^{(t+1)}$
5:     Each selected client $k \in \mathcal{M}^{(t)}$ sends $\boldsymbol{\theta}_k^{(t+1)}$ back to the server
6:     Server aggregates the $\boldsymbol{\theta}$'s as $\boldsymbol{\theta}^{(t+1)} := \sum_{k \in \mathcal{M}^{(t)}} \frac{N_k}{\sum_{k \in \mathcal{M}^{(t)}} N_k} \boldsymbol{\theta}_k^{(t+1)}$
7: **end for**

---

in FL, we refer interested readers to (Kairouz et al., 2019). Unlike these works that are agnostic to the existence of CL periods, we design a novel CL periods-aware FL framework. Importantly, we remark that our CL periods-aware FL framework, FedCL is *orthogonal* to these methods, since FedCL *merely* augments a state-of-the-art FL method to adaptively determine the number of clients that participate in each FL training round, rather than changing the way how the FL method selects clients.

## 3 BACKGROUND

**The Federated Optimization Setting.** Consider the federated architecture where $M$ clients jointly solve the optimization problem: $\min_{\boldsymbol{\theta} \in \mathbb{R}^d} F(\boldsymbol{\theta}) := \sum_{k=1}^{M} p_k F_k(\boldsymbol{\theta})$, where $p_k = N_k/N$ represents the relative sample size, and $F_k(\boldsymbol{\theta}) = \frac{1}{N_k} \sum_{\xi \in \mathcal{D}_k} \ell_k(\boldsymbol{\theta}; \xi)$ is the local objective function at the $k$-th client. Here $\ell_k$ denotes the loss function defined by the learning model, $\xi$ represents a data sample from local dataset $\mathcal{D}_k$, and $\mathcal{M}$ denotes the set of clients.

**The Federated Averaging Algorithm.** *Federated Averaging* (FedAvg) (McMahan et al., 2017) is the first and most common algorithm used to solve the above optimization problem through aggregating the locally trained models at the central server at the end of each communication round. At the initial step, the central server randomly initializes a global model $\boldsymbol{\theta}^{(0)}$. At each round, a *fixed* number of randomly selected clients run $E$ iterations of local solver, e.g., the stochastic gradient descent (SGD) (Yu et al., 2019; Wang & Joshi, 2019; 2021), and then the resulting model updates are averaged. The details of FedAvg are summarized in Algorithm 1, where $\mathcal{M}^{(t)} \subseteq \mathcal{M}$ and $m := |\mathcal{M}^{(t)}| \leq M, \forall t$.

Although the performance of FedAvg has been improved in both theory and practice by recent literature such as FedProx (Li et al., 2020a), FedNova (Wang et al., 2020c), SCAFFOLD (Karimireddy et al., 2020), VRL-SGD (Liang et al., 2019), FedBoost (Hamer et al., 2020), FedMA (Wang et al., 2020b) and FetchSGD (Rothchild et al., 2020), FedAvg is the first and the most widely used one. As a result, we see FedAvg as our basic block. *Our proposed CL periods-aware FL framework FedCL is orthogonal to and can be easily combined with these methods (see Section 5).* Moreover, FedCL is also compatible with and complementary to other techniques such as gradient compression/quantization (Basu et al., 2019; Haddadpour et al., 2021).

**CL Periods in FL.** The latest series of studies have identified *CL periods* or the initial training phase that are important for training a high-quality model in both centralized training (Achille et al., 2019; Jastrzebski et al., 2019) and decentralized training (Yan et al., 2022). We build upon recent work by (Yan et al., 2022), who showed that if the training dataset for each client is not recovered to the entire training dataset early enough in the training process, the testing accuracy of FL is permanently impaired. We extend these ideas to aid in the design of adaptive client selection in FL and show that more clients are only required during these CL periods.

## 4 FEDCL: A CL PERIODS-AWARE CLIENT SELECTION FRAMEWORK

This section describes our proposed approach to efficiently detect CL periods in federated settings that lay out the rationale behind our method. The rest of this section focuses on our proposed framework FedCL that augments client selection in state-of-the-art methods with CL periods.

Figure 1: Comparison of detecting CL periods in federated settings using FGN with $\delta = 0.01$ and FedFIM, where the shade and double-arrows indicate identified CL periods. The results are conducted using AlexNet on (a) CIFAR-10 and (b) Fashion-MNIST datasets, which are non-IID partitioned across 128 clients using Dirichlet distributions $\text{Dir}_{128}(0.1)$, $\text{Dir}_{128}(0.2)$, and $\text{Dir}_{128}(0.3)$, respectively.

## 4.1 ADAPTIVE CLIENT SELECTION VIA CL PERIODS AWARENESS

(Yan et al., 2022) showed that the final test accuracy of FL is dramatically affected by early training phases. They setup experiments where only partial datasets are available for the first few communication rounds and then continue training the model with entire training datasets for the rest of communication rounds. Surprisingly, the FL model trained in this way showed a permanent impaired test accuracy performance no matter how many additional training rounds are performed after CL periods. We extend these ideas to aid in the design of adaptive client selection in FL since it is a major trend of improving FL efficiency and handling heterogeneity. As motivated by aforementioned works, it is clear that finding an adaptive client selection scheme is akin to finding CL periods in FL training process. To this end, we begin with how to efficiently detect CL periods in FL training process.

**Detecting Critical Learning Periods.** Prior works use the changes in eigenvalues of the Hessian or approximating the Hessian using (federated) Fisher information (Achille et al., 2019; Jastrzebski et al., 2019; Yan et al., 2022) as an indicator to detect CL periods. We deviate from these works and develop an approach based on federated gradient norm (FGN), which can be efficiently computed.

Considering the difference in training loss for an individual data sample $\xi$, let $g(\boldsymbol{\theta}; \xi) = \frac{\partial}{\partial \theta} \ell(\boldsymbol{\theta}; \xi)$ denote the gradient of the loss function evaluated on $\xi$. After performing a step SGD on this sample, the training loss $\Delta \ell = \ell(\boldsymbol{\theta} - \eta g(\boldsymbol{\theta}; \xi); \xi) - \ell(\boldsymbol{\theta}; \xi)$ can be approximated by its gradient norm using Taylor expansion, i.e., $\Delta \ell \approx -\eta \|g(\boldsymbol{\theta}; \xi)\|^2$. As a result, the overall training loss at the $t$-th round, which we define as the FGN, can be approximated using the weighted average of training loss across all selected clients, i.e.,

$$\text{FGN}(t) = \sum_{k \in \mathcal{M}^{(t)}} \frac{N_k}{\sum_{k \in \mathcal{M}^{(t)}} N_k} \Delta \ell_k^{(t)}. \tag{1}$$

We compare the CL periods identified by our FGN approach with the federated Fisher information (FedFIM) approach in (Yan et al., 2022). From Figure 1, we observe that these two approaches yield similar results, but our FGN approach is much more computationally efficient (being orders of magnitude faster to compute) and can be easily leveraged for client selection during the training process in an online manner. For example, the FedFIM approach takes up to $9\times$ more computation time and consumes $40\times$ more memory than our FGN approach (See Appendix A.2.1 for details).

## 4.2 THE DESIGN OF FEDCL FRAMEWORK

We now describe FedCL, our proposed framework that adaptively determines the number of selected clients for FL training by leveraging identified CL periods. Again, we use FedAvg as the building block, and our framework can be easily combined with other existing methods, which we will illustrate in Section 5. Our FedCL builds on the identified CL periods, which can be efficiently detected by FGN as discussed previously (see Figure 1). To this end, we develop a simple *threshold-based rule* to detect the CL periods based on FGN as follows: if

$$\frac{\text{FGN}(t) - \text{FGN}(t-1)}{\text{FGN}(t-1)} \geq \delta, \tag{2}$$

then the current round $t$ is in CL periods, where $\delta$ is the threshold used to declare CL periods. We set $\delta = 0.01$ as the default value in our experiments and will investigate its impact in Section 5.

---

**Algorithm 2** FedCL: A CL Periods-Aware Client Selection Framework

---

**Input:** $\mathcal{M}, \eta, E, T, \delta, m$ selected clients with initial global model $\boldsymbol{\theta}^{(0)}$

1: **for** $t = 0, 1, \cdots, T - 1$ **do**
2:    Server selects a subset $\mathcal{M}^{(t)}$ of $\mathcal{M}$ clients at random
3:    **if** $\frac{\text{FGN}(t) - \text{FGN}(t-1)}{\text{FGN}(t-1)} \geq \delta$ **then**
4:      Clients in $\mathcal{M}^{(t)}$ update the local model and server aggregates local models via FedAvg
5:      $|\mathcal{M}^{(t+1)}| \leftarrow \min\{2|\mathcal{M}^{(t)}|, M\}$ //Doubling the clients in CL periods
6:    **else**
7:      $|\mathcal{M}^{(t+1)}| \leftarrow \max\{\frac{1}{2}|\mathcal{M}^{(t)}|, \frac{1}{2}m\}$ //Halving the clients after CL periods
8:    **end if**
9: **end for**

---

Per our discussions on CL periods, the final model accuracy will be permanently impaired if not enough clients are involved in CL periods no matter how much additional training is performed after the period (Yan et al., 2022). Therefore, our FedCL framework increases the number of selected clients of FedAvg from $n_0$ to $2n_0$, implying that more clients now participate in improving the global model in the next round during the CL periods. Using the model learned from the previous round $\boldsymbol{\theta}_{n_0}$ as the initial model, the $2n_0$ selected clients employ the FedAvg and continue the learning procedure to reach a global model $\boldsymbol{\theta}_{2n_0}$. The procedure of geometrically increasing the number of selected clients continues till the set of selected clients contains all the available $M$ clients when the communication rounds are still in CL periods. Since the final accuracy of using partial datasets is similar to that of using all dataset after CL periods (Yan et al., 2022), FedCL starts to gradually decrease the number of selected clients after CL periods for the sake of communication efficiency. Algorithm 2 summarizes the CL periods-aware FedAvg algorithm, dubbed as FedCL.

From a high-level perspective, FedCL exploits more clients in the initial phase of the learning procedure than a fixed number of clients for FedAvg in each round, to promptly reach a global model with higher accuracy since the initial learning phase plays a critical role in FL performance. By doing so, we hypothesize that the SGD is navigating to the steeper parts of the loss surface of the global model during CL periods since a larger amount of data samples have contributed to the global model. However, the communication overhead of such an approach is relatively large since more clients are involved in FL training in each communication round. By gradually decreasing the number of selected clients after CL periods, the communication overhead of FedCL improves without hurting the final model accuracy. The key point is that more clients join the training process in the initial learning phase, and only a smaller number of clients are needed after CL periods. As a result, FedCL consistently improves the model accuracy while maintains comparable or even better communication efficiency than FedAvg.

As the proposed FedCL in Algorithm 2 provides a general framework to augment client selection with identified CL periods in federated settings, one needs to specify the inner optimization subroutine (e.g., line 4 in Algorithm 2) to quantify the improvement of the proposed approach. In particular, we set the subroutine to be FedAvg in Algorithm 2 since it is the most common algorithm and the building block of many variants in federated settings. This subroutine could be any federated learning algorithms (with possible variants), such as FedProx (Li et al., 2020a), VRL-SGD (Liang et al., 2019) and FedNova (Wang et al., 2020c), which we will numerically illustrate in Section 5.

## 5 EXPERIMENTS

We first present an empirical study of FedCL in Section 5.2, and then illustrate the generalization of our proposed framework by combining CL periods with other methods such as FedProx, VRL-SGD and FedNova in Section 5.3. We study classification problems using two representative DNN models: AlexNet (Krizhevsky et al., 2012) and VGG-11 (Simonyan & Zisserman, 2015) on non-IID partitioned CIFAR-10 and CIFAR-100 (Krizhevsky et al., 2009), and Fashion-MNIST (Xiao et al., 2017) datasets. We further investigate the task of next-character prediction on the dataset of *The Complete Works of William Shakespeare* (Shakespeare) (McMahan et al., 2017). We relegate details of datasets and models, and additional experimental results, particularly on CIFAR-100 and Shakespeare, to Appendix A.

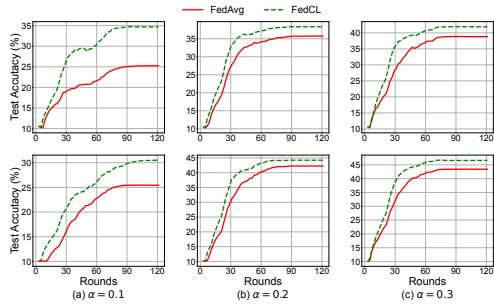

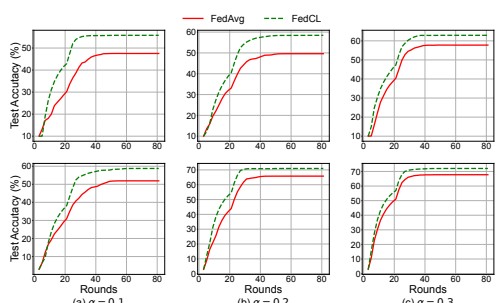

Figure 2: Test accuracy of **FedAvg** and **FedCL** using **(top) AlexNet** and **(bottom) VGG-11** on non-IID **CIFAR-10**.

Figure 3: Test accuracy of **FedAvg** and **FedCL** using **(top) AlexNet** and **(bottom) VGG-11** on non-IID **Fashion-MNIST**.

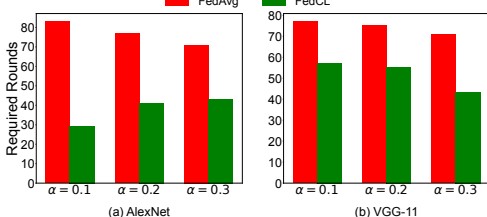

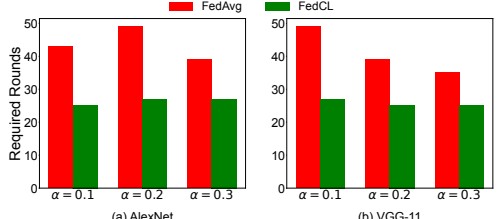

Figure 4: Communication efficiency of **FedAvg** and **FedCL** on non-IID **CIFAR-10**.

Figure 5: Communication efficiency of **FedAvg** and **FedCL** on non-IID **Fashion-MNIST**.

## 5.1 EXPERIMENT SETUP

We implement FedCL and considered baselines in PyTorch (Paszke et al., 2017) on Python 3 with three NVIDIA RTX A6000 GPUs. For CIFAR-10, CIFAR-100 and Fashion-MNIST datasets, we simulate the non-IID FL scenario by considering a heterogeneous partition for which the number of data points and class proportions are unbalanced. In particular, we simulate a heterogeneous partition into $M$ clients by sampling $\boldsymbol{p}_k \sim \text{Dir}_M(\alpha)$, where $\alpha$ is the parameter of the Dirichlet distribution. We choose $\alpha = 0.1, 0.2, 0.3$ in our experiments as done in (Wang et al., 2020b;c). The level of heterogeneity among local datasets across different clients can be reduced when $\alpha$ increases. We consider the total number of clients to be 128. The local learning rate $\eta$ is initialized as $0.01$ and decayed by a constant factor after each communication round. We set the weight decay to be $10^{-5}$. The detection threshold is $\delta = 0.01$ and the number of local training epochs is $E = 2$. An ablation study is conducted in Section 5.4 to investigate the impact of these hyperparameters. We run each experiment with three independent trials and report the mean results.

## 5.2 IMPORTANCE OF CL PERIODS AWARENESS: FEDCL *v.s.* FEDAVG

In this experiment, we study the performance of FedCL with accuracy and communication efficiency. Our goal is to compare FedCL to FedAvg in terms of the final test accuracy and the number of communication rounds needed for the global model to achieve good performance on the test data.

**Test Accuracy.** The test accuracy comparisons on non-IID partitioned CIFAR-10 and Fashion-MNIST with FedAvg selecting 16 clients in each round are shown in Figures 2 and 3, respectively. Obviously, FedCL consistently outperforms FedAvg in all scenarios with an improved final test accuracy up to 11%. Its advantage is especially pronounced when the dataset is partitioned across clients using a Dirichlet distribution with parameter 0.1, i.e., the datasets across clients are highly non-IID. Not surprisingly, we observe the importance of CL periods awareness in training efficiency, which is fully reflected via the test accuracy. For example in Figures 2(a) and 3(a), FedCL exhibits a dramatic accuracy increase in the early phase of the FL training process. This coincides with the fact that FedCL selects a larger number of clients in each round in the early phase due to the awareness of CL periods (lines 3-6 in Algorithm 2). Though the accuracy slightly decreases in a short period due to the decreased number of selected clients (lines 7-9 in Algorithm 2), the final test accuracy is significantly improved. Our findings on the importance of CL periods awareness in the FL training

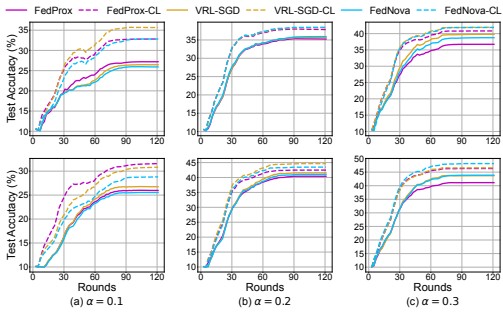

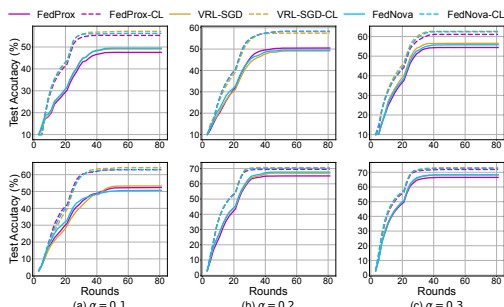

Figure 6: Test accuracy of **FedProx, VRL-SGD, FedNova** and **FedProx-CL, VRL-SGD-CL, FedNova-CL** using **(top)** AlexNet and **(bottom)** VGG-11 on non-IID **CIFAR-10**.

Figure 7: Test accuracy of **FedProx, VRL-SGD, FedNova** and **FedProx-CL, VRL-SGD-CL, FedNova-CL** using **(top)** AlexNet and **(bottom)** VGG-11 on non-IID **Fashion-MNIST**.

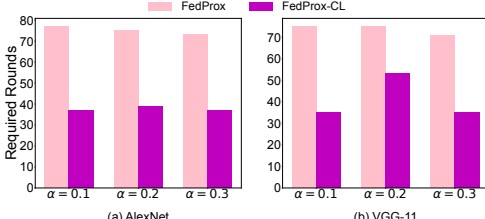

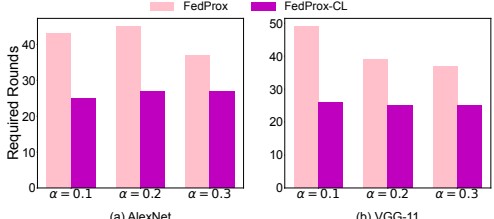

Figure 8: Communication of **FedProx** and **FedProx-CL** on non-IID **CIFAR-10**.

Figure 9: Communication of **FedProx** and **FedProx-CL** on non-IID **Fashion-MNIST**.

process, e.g., for client selection, seem to be consistent with recently reported observations that the initial learning phase plays a key role in determining the outcome of the training process.

**Communication Efficiency.** The benefit of CL periods awareness for FL training is further reflected via communication efficiency. In Figures 4 and 5, we report the communication rounds required for FedCL and FedAvg to achieve a targeted accuracy, which is chosen to be the final test accuracy of FedAvg as reported in Figures 2 and 3, respectively. Comparisons on other targeted test accuracy can be found in Appendix A.2. It is clear that FedCL requires fewer rounds to achieve the same test accuracy. Again this advantage is pronounced on highly non-IID data partitions. We further compute the average number of clients involved in each round for FedCL and FedAvg to achieve its final test accuracy. We observe that FedCL consistently improves the model accuracy (as shown Figures 2 and 3) with its average number of clients involved in each round being $0.95\times$ to $1.1\times$ that of FedAvg. This is due to the fact that more clients are only needed during CL periods and a smaller number of clients are need afterwards in our FedCL framework.

## 5.3 GENERALIZATION

As mentioned earlier, our proposed CL periods-aware FL framework, FedCL is orthogonal to existing state-of-the-art methods, and hence can be easily combined with these methods by simply replacing the inner optimization subroutine (FedAvg) in Algorithm 2 with the corresponding methods. To this end, we study the generalization of FedCL and consider three state of the arts, i.e., FedProx (Li et al., 2020a), VRL-SGD (Liang et al., 2019) and FedNova (Wang et al., 2020c). We call the corresponding CL periods-aware methods as FedProx-CL, VRL-SGD-CL, and FedNova-CL, respectively.

We notice that the performance of FedProx depends on the hyperparameter $\mu$, i.e., the coefficient associated with the proximal term of each local objective. We tune this parameter using grid search and report the best value of $\mu = 0.01$ for AlexNet experiments and $\mu = 0.001$ for VGG-11 experiments. We present results on non-IID datasets across 128 clients with FedProx, VRL-SGD and FedNova selecting 16 clients in each iteration. Again, the CL periods awareness significantly improves the test performance of baseline methods, i.e., FedProx-CL,VRL-SGD-CL and FedNova-CL outperforms FedProx, VRL-SGD and FedNova, respectively, in all scenarios with an improved final test accuracy up to 11%, 13% and 10%, respectively, as shown in Figures 6 and 7. Its advantage is

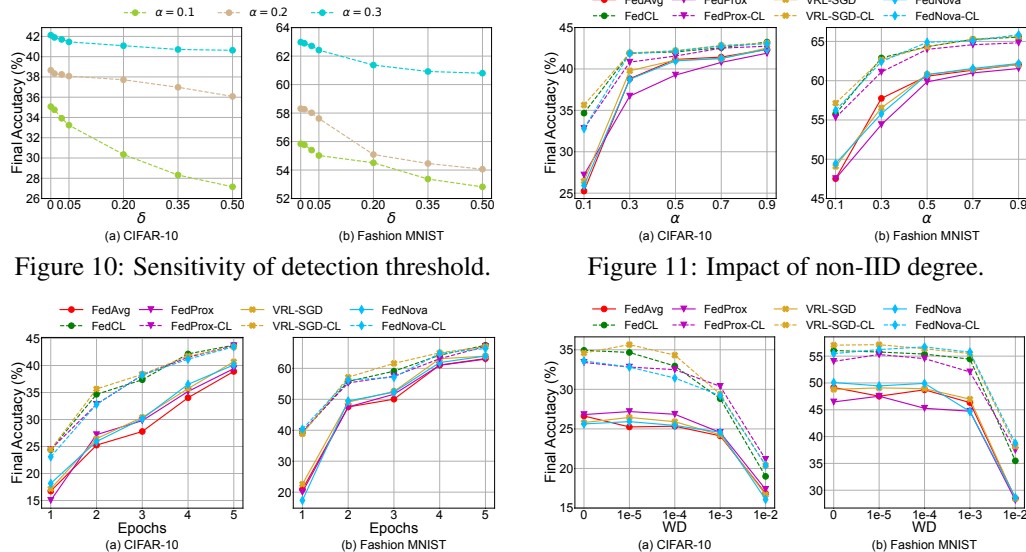

Figure 10: Sensitivity of detection threshold.

Figure 11: Impact of non-IID degree.

Figure 12: Effect of local training epochs.

Figure 13: Effect of weight decays.

especially pronounced on highly non-IID dataset across clients in Figures 6(a) and 7(a). Likewise, the CL periods augmented method, e.g., FedProx-CL requires fewer rounds to achieve the final test accuracy of the corresponding baseline FedProx, as shown in Figures 8 and 9, while maintaining a comparable average number of clients involved in each round. Similar observations can be made for VRL-SGD-CL vs. VRL-SGD and FedNova-CL vs. FedNova, which can be found in Appendix A.2.

## 5.4 ABLATION STUDY

**Detection Thresholds.** As discussed previously in Figure 1, our experiments reveal that CL periods can be efficiently identified using the easy-to-compute FGN via a simple threshold-type rule in Equation (2). We now evaluate the sensitivity of the threshold value $\delta$ used to declare CL periods. The candidate threshold values we consider are $\{0, 0.01, 0.03, 0.05, 0.2, 0.35, 0.5\}$, and the corresponding final test accuracy of FedCL using AlexNet on non-IID partitioned CIFAR-10 and Fashion-MNIST is illustrated in Figure 10. When data partitions are highly non-IID (i.e., $\alpha = 0.1$), the CL periods declaration determined by $\delta$ has an observable effect on the final accuracy. This is because as $\delta$ becomes larger, fewer rounds in the initial phase are declared as CL periods by Equation (2). As a result, the effect of CL periods awareness on the final test accuracy is shallowed since FedCL only uses a larger number of clients in fewer rounds compared to FedAvg according to Algorithm 2. On the other hand, our CL periods-aware framework FedCL is robust to the detection process, i.e., tolerant to detection errors with different threshold values when data partitions are not highly non-IID. Similar observations can be made for FedProx-CL,VRL-SGD-CL and FedNova-CL and hence are relegated to Appendix A.2. To this end, we set $\delta = 0.01$ in all of our experiments.

**Non-IID Degree.** We simulate a heterogeneous data partition into $M$ clients using the Dirichlet distribution with parameter $\alpha$. From Figure 11, we observe that the CL periods awareness consistently improves the final test accuracy of a state-of-the-art method across all values of $\alpha$ in consideration. For example, FedCL always outperforms FedAvg, and FedProx-CL always outperforms FedProx. The benefits of CL periods awareness are especially pronounced when the datasets across clients are highly non-IID (i.e., a smaller value of $\alpha$). Hence, we choose $\alpha = 0.1, 0.2, 0.3$ for illustrations in above experiments. For ease of readability, we set $\alpha = 0.1$ in the rest of ablation studies and relegate results on $\alpha = 0.2, 0.3$ to Appendix A.2. In addition, as the non-IID degree decreases (as $\alpha$ increases), the final test accuracy of FedCL, FedProx-CL, VRL-SGD-CL and FedNova-CL increases. This is consistent with recently reported observations, e.g., in (Lin et al., 2020; Achituve et al., 2021; Gong et al., 2021) that non-IID degree degrades the model final test accuracy.

**Local Training Epochs.** We note that the number of local training epochs (denoted $E$) is a common parameter shared by considered baselines, which reportedly has an impact on the performance of FedAvg (McMahan et al., 2017; Wang et al., 2020b). To this end, we evaluate the effect of $E$ using

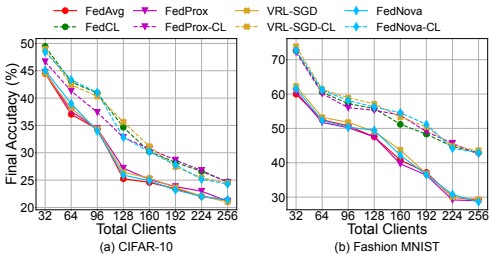
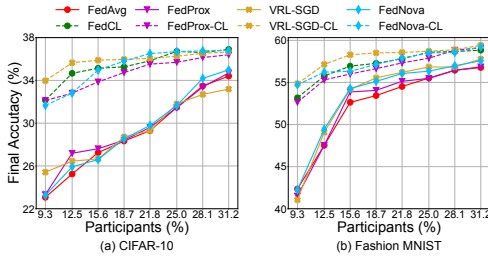

Figure 14: Impact of number of clients.   Figure 15: Impact of client participation rate.

AlexNet on non-IID partitioned CIFAR-10 and Fashion-MNIST with $\alpha = 0.1$. The candidate local epochs we consider are $E \in \{1, 2, 3, 4, 5\}$ as done in (Wang et al., 2020c). From Figure 12, we observe that increasing the number of local epochs improves the final test accuracy in general, and the CL periods awareness consistently improves the final test accuracy of state-of-the-art methods across all values of $E$. Since the gains in test accuracy exhibit the "diminishing return effect" as the number of local epochs increases, we set $E = 2$ in all of our experiments.

**Weight Decay.** Though the CL periods in FL are robust to the values of weight decays as reported in (Yan et al., 2022), the final test accuracy using AlexNet on non-IID partitioned CIFAR-10 and Fashion-MNIST with $\alpha = 0.1$ is still affected with the values of weight decays as shown in Figure 13. Again, we consistently observe the benefits of CL periods awareness across all values of weight decays. Since the advantage decreases as the weight decay increases, we set the weight decay to be $10^{-5}$ in our experiments.

**Number of Clients.** In all of our above experiments, we consider a FL setting with 128 clients in total. We now consider the same experimental settings as above besides varying the total number of clients in the system using AlexNet on non-IID partitioned CIFAR-10 and Fashion-MNIST with $\alpha = 0.1$. As shown in Figure 14, the advantage of CL periods awareness exhibits across all settings, i.e., FedCL (resp. FedProx-CL) outperforms FedAvg (resp. FedProx) regardless of the total number of clients. Without loss of generality, we choose $M = 128$ in all of our experiments.

**Client Participation Rate.** In all of our experiments, FedAvg, FedProx, VRL-SGD and FedNova select 16 out of 128 clients to participate in each training round, i.e., the participation rate is 12.5%. We now investigate the impact of client participation rates on the model accuracy and the awareness of CL periods using AlexNet on non-IID partitioned CIFAR-10 and Fashion-MNIST with $\alpha = 0.1$. Again, when a state-of-the-art method is augmented with the CL periods, the final test accuracy is consistently improved across all client participation rates. The advantage is particularly pronounced with a low participation rate. This is quite intuitive since in our CL periods aware framework, FedCL selects more clients during the CL periods than FedAvg (see line 5 in Algorithm 2), and hence the benefits are more obvious when FedAvg has a low client participation rate. We select 16 clients, i.e., a 12.5% participation rate for all state-of-the-art methods via considering the tradeoff between final test accuracy and benefits of CL periods awareness.

## 6 Conclusion

In this paper, we presented FedCL, a simple but powerful CL periods-aware FL framework for adaptive client selection. FedCL worked by adaptively choosing more clients in CL periods during the FL training process and fewer clients elsewhere. We proposed a practical and easy-to-compute federated gradient norm (FGN) metric to identify such CL periods during the training process in an online manner. We showed that FedCL significantly improved the final test accuracy by up to 11% compared to its counterpart FedAvg using different models and datasets, while maintaining comparable or even better communication efficiency. Finally, we illustrated the generalization of our proposed CL periods aware framework via manipulating the client selection of state-of-the-art methods augmented by FedCL. In the future work, we want to extend FedCL to improve FL of different machine learning models on other popular techniques such as gradient compression/quantization, fair aggregation, personalization, and adversarial attacks. We also believe that it is important to study the performance of FedCL on other models and datasets.

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

# A    EXTENSIVE REVIEW AND RESULTS OF EXPERIMENTS

We provide details of datasets and models in Appendix A.1 and additional experimental results in Appendix A.2.

| Parameter | Shape | Layer hyper-parameter |
|---|---|---|
| layer1.conv1.weight | $3 \times 64 \times 3 \times 3$ | stride:1; padding: 1 |
| layer1.conv1.bias | 64 | N/A |
| pooling.max | N/A | kernel size:2; stride: 2 |
| layer2.conv2.weight | $64 \times 128 \times 3 \times 3$ | stride:1; padding: 1 |
| layer2.conv2.bias | 128 | N/A |
| pooling.max | N/A | kernel size:2; stride: 2 |
| layer3.conv3.weight | $128 \times 256 \times 3 \times 3$ | stride:1; padding: 1 |
| layer3.conv3.bias | 256 | N/A |
| layer4.conv4.weight | $256 \times 256 \times 3 \times 3$ | stride:1; padding: 1 |
| layer4.conv4.bias | 256 | N/A |
| pooling.max | N/A | kernel size:2; stride: 2 |
| layer5.conv5.weight | $256 \times 512 \times 3 \times 3$ | stride:1; padding: 1 |
| layer5.conv5.bias | 512 | N/A |
| layer6.conv6.weight | $512 \times 512 \times 3 \times 3$ | stride:1; padding: 1 |
| layer6.conv6.bias | 512 | N/A |
| pooling.max | N/A | kernel size:2; stride: 2 |
| layer7.conv7.weight | $512 \times 512 \times 3 \times 3$ | stride:1; padding: 1 |
| layer7.conv7.bias | 512 | N/A |
| layer8.conv8.weight | $512 \times 512 \times 3 \times 3$ | stride:1; padding: 1 |
| layer8.conv8.bias | 512 | N/A |
| pooling.max | N/A | kernel size:2; stride: 2 |
| dropout | N/A | p=20% |
| layer9.fc9.weight | $4096 \times 512$ | N/A |
| layer9.fc9.bias | 512 | N/A |
| layer10.fc10.weight | $512 \times 512$ | N/A |
| layer10.fc10.bias | 512 | N/A |
| dropout | N/A | p=20% |
| layer11.fc11.weight | $512 \times 10$ | N/A |
| layer11.fc11.bias | 10 | N/A |

Table 1: Detailed information of the VGG-11 architecture used in our experiments. All non-linear activation function in this architecture is ReLU. The shapes for convolution layers follow $(C_{in}, C_{out}, c, c)$.

## A.1    SUMMARY OF DATASETS AND MODEL ARCHITECTURES

We implement FedCL and considered baselines in PyTorch (Paszke et al., 2017) on Python 3 with three NVIDIA RTX A6000 GPUs, 48GB with 128GB RAM. We conduct experiments on the popular datasets CIFAR-10 and CIFAR-100 (Krizhevsky et al., 2009), Fashion-MNIST (Xiao et al., 2017) and Shakespeare (McMahan et al., 2017). The CIFAR-10 and CIFAR1-00 dataset consists of 60,000 32×32 color images in 10 and 100 classes, respectively, where 50,000 samples are for training and the other 10,000 samples for testing. The Fashion-MNIST datasets contain handwritten digits with 60,000 samples for training and 10,000 samples for testing, where each sample is an 28×28 grayscale images over 10 classes. The Shakespeare dataset consists of 74 characters with 734,057 training data and 70,657 testing data. We simulate a heterogeneous partition into $M$ clients by sampling $\boldsymbol{p}_k \sim \text{Dir}_M(\alpha)$, where $\alpha$ is the parameter of the Dirichlet distribution (Wang et al., 2020b;c). Specifically, for each class of samples, set the class probability in each client by sampling from a Dirichlet distribution with the same $\alpha$ parameter. For instance, when $\alpha = 0.5$, sampling $p_k \sim \text{Dir}(0.5)$ and allocating a $p_{k,j}$ proportion of the training instances of class $k$ to local client $j$.

We summarize the details of VGG-11 (Simonyan & Zisserman, 2015) and AlexNet (Krizhevsky et al., 2012) architectures used in our experiments for classification tasks in Tables 1 and 2, respectively.

For the language task, we train a stacked character-level LSTM language model as in (Kim et al., 2016; McMahan et al., 2017), which is summarized in Table 3

| Parameter | Shape | Layer hyper-parameter |
|---|---|---|
| layer1.conv1.weight | $3 \times 64 \times 3 \times 3$ | stride:2; padding: 1 |
| layer1.conv1.bias | 32 | N/A |
| pooling.max | N/A | kernel size:2; stride: 2 |
| layer2.conv2.weight | $64 \times 192 \times 3 \times 3$ | stride:1; padding: 1 |
| layer2.conv2.bias | 64 | N/A |
| pooling.max | N/A | kernel size:2; stride: 2 |
| layer3.conv3.weight | $192 \times 384 \times 3 \times 3$ | stride:1; padding: 1 |
| layer3.conv3.bias | 128 | N/A |
| layer4.conv4.weight | $384 \times 256 \times 3 \times 3$ | stride:1; padding: 1 |
| layer4.conv4.bias | 128 | N/A |
| layer5.conv5.weight | $256 \times 256 \times 3 \times 3$ | stride:1; padding: 1 |
| layer5.conv5.bias | 256 | N/A |
| pooling.max | N/A | kernel size:2; stride: 2 |
| dropout | N/A | p=5% |
| layer6.fc6.weight | $1024 \times 4096$ | N/A |
| layer6.fc6.bias | 512 | N/A |
| dropout | N/A | p=5% |
| layer7.fc7.weight | $4096 \times 4096$ | N/A |
| layer7.fc7.bias | 512 | N/A |
| layer8.fc8.weight | $4096 \times 10$ | N/A |
| layer8.fc8.bias | 10 | N/A |

Table 2: Detailed information of the AlexNet architecture used in our experiments. All non-linear activation function in this architecture is ReLU. The shapes for convolution layers follow $(C_{in}, C_{out}, c, c)$.

| Parameter | Shape | Layer hyper-parameter |
|---|---|---|
| layer1.embeding | $80 \times 256$ | N/A |
| layer2.lstm | $256 \times 512$ | num_layers=2, batch_first=True |
| dropout | N/A | p=5% |
| layer3.fc.weight | $512 \times 80$ | N/A |
| layer3.fc.bias | 80 | N/A |

Table 3: Detailed information of the LSTM architecture used in our experiments.

## A.2 Additional Experimental Results

Results complementary to Section 5 on non-IID partitioned CIFAR-10 and Fashion-MNIST datasets are given in Appendix A.2.2. Experimental results on non-IID partitioned CIFAR-100 and Shakespeare datasets are presented in Appendix A.2.3.

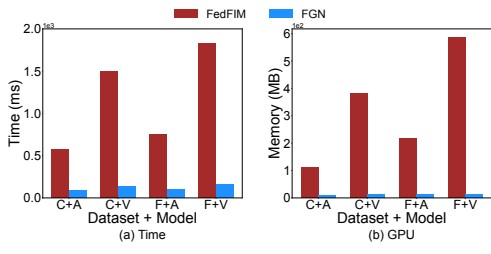

Figure 16: Computation time and memory consumption of FGN and FedFIM approaches to detect CL periods.

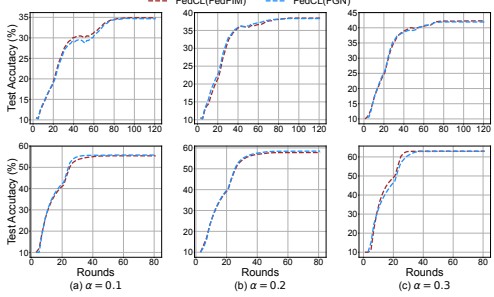

Figure 17: Test accuracy of **FedCL** using FGN and FedFIM approaches to detect CL periods.

### A.2.1 COMPARISON BETWEEN FGN AND FEDFIM BASED APPROACH TO DETECT CL PERIODS

As discussed in Section 4, we propose a lightweight FGN based approach to detect CL periods that can be easily leveraged for client selection during the training process in an online manner. Though the detection performance is similar between FGN and FedFIM approach as shown in Figure 1, our FGN approach is much more computationally efficient. For example, consider the settings in Section 4, the computation time and memory consumption of FGN and FedFIM under the settings in Appendix A.1 are presented in Figure 16, where C+A, C+V, M+F, and F+A represent AlexNet on CIFAR-10, VGG-11 on CIFAR-10, FC on MNIST, and AlexNet on Fashion-MNIST, respectively. Since the identified CL periods by using FGN are almost the same as those identified by using FedFIM (see Figure 1), the test accuracy of FedCL is expected to be the same. For instance, the test accuracy of FedCL when leveraging the CL periods identified by FGN and FedFIM, which we denote as FedCL (FGN) and FedCL (FedFIM), respectively, using AlexNet on CIFAR-10 is reported as Figure 17.

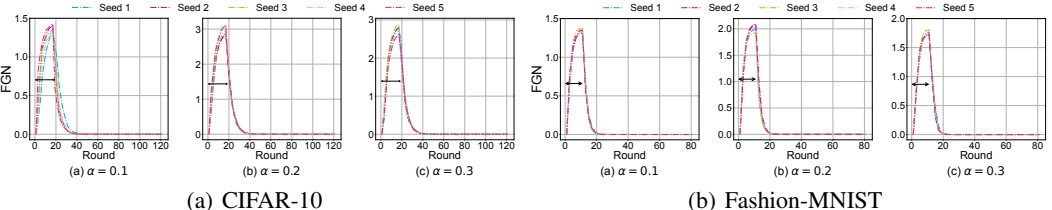

Figure 18: The impact of random seed when using FGN to detect CL periods. The results are conducted using AlexNet on (a) CIFAR-10 and (b) Fashion-MNIST datasets, which are non-IID partitioned across 128 clients using Dirichlet distributions $\mathrm{Dir}_{128}(0.1)$, $\mathrm{Dir}_{128}(0.2)$, and $\mathrm{Dir}_{128}(0.3)$, respectively.

We further evaluate the impact of random seed when using FGN to detect CL periods. In particular, we randomly generate five seeds, and report the identified CL periods in Figure 18. We observe that our FGN can consistently identify the CL periods across different random seeds and the identified CL periods are almost the same. This performance is further pronounced when we compare the test accuracy as shown in Figure 19. These results further support the robustness of our proposed FGN metric.

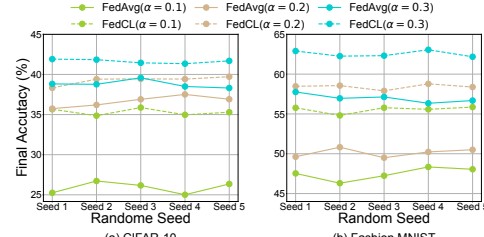

Figure 19: The test accuracy of FedAvg and FedCL using different random seeds.

### A.2.2 ADDITIONAL RESULTS ON CIFAR-10 AND FASHION-MNIST DATASETS

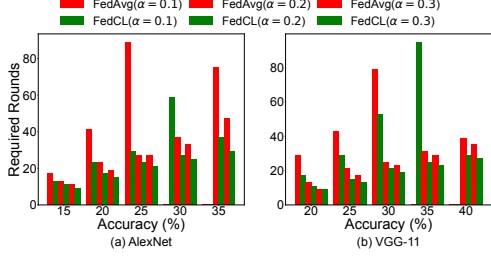
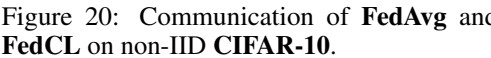

Figure 20: Communication of **FedAvg** and **FedCL** on non-IID **CIFAR-10**.

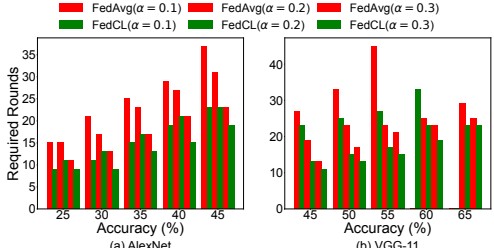

Figure 21: Communication of **FedAvg** and **FedCL** on non-IID **Fashion-MNIST**.

**Communication Efficiency.** Similar to Figures 4 and 5, we report the communication rounds required for FedCL and FedAvg to achieve some common targeted accuracies in Figures 20 and 21. Again, we observe that FedCL requires fewer rounds to achieve the same test accuracy as FedAvg. The communication comparison for FedProx, VRL-SGD and FedNova that is similar to Figures 8 and 9, are presented in Figures 22 and 23 for VRL-SGD vs. VRL-SGD-CL, and in Figures 24 and 25 for FedNova vs. FedNova-CL.

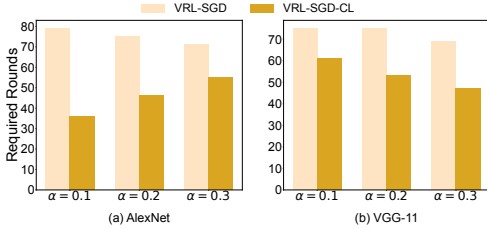

Figure 22: Communication of **VRL-SGD** and **VRL-SGD-CL** on non-IID **CIFAR-10**.

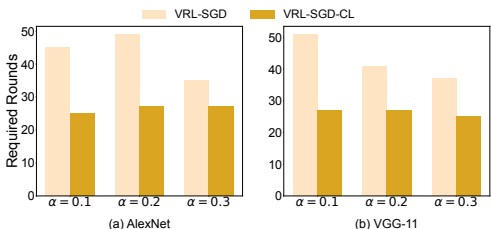

Figure 23: Communication of **VRL-SGD** and **VRL-SGD-CL** on non-IID **Fashion-MNIST**.

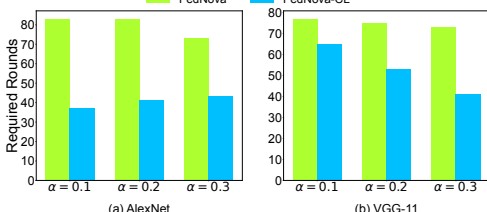

Figure 24: Communication of **FedNova** and **FedNova-CL** on non-IID **CIFAR-10**.

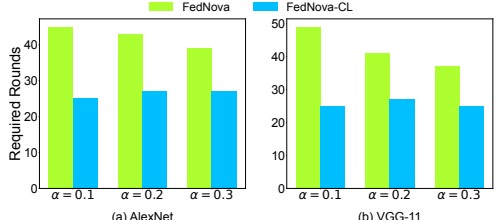

Figure 25: Communication of **FedNova** and **FedNova-CL** on non-IID **Fashion-MNIST**.

**Detection Thresholds.** Complementary to the results shown in Figure 10, the detection thresholds also have the same impact on the final test accuracy of FedProx-CL, VRL-SGD-CL and FedNova-CL using AlexNet on non-IID partitioned CIFAR-10 and Fashion-MNIST as illustrated in Figures 26, 27 and 28.

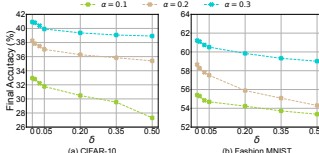

Figure 26: Sensitivity of detection threshold: FedProx-CL.

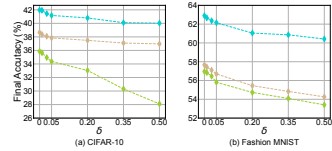

Figure 27: Sensitivity of detection threshold: VRL-SGD-CL.

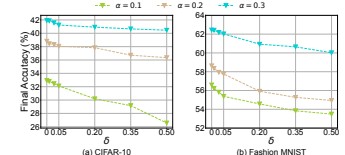

Figure 28: Sensitivity of detection threshold: FedNova-CL.

### A.2.3 RESULTS ON CIFAR-100 AND SHAKESPEARE DATASETS

**Test Accuracy.** We present the test accuracy on non-IID partitioned CIFAR-100 and Shakespeare datasets in Figures 29 and 30, and Figures 31 and 32, respectively. Again we observe that the CL periods awareness consistently improves the final test accuracy.

**Communication Efficiency.** The comparisons of communication rounds required for FedCL, FedProx-CL, VRL-SGD-CL and FedNova-CL to achieve the final test accuracy of FedAvg, FedProx, VRL-SGD and FedNova, respectively, are presented in Figures 33, 34, 35 and 36, respectively, using the non-IID partitioned CIFAR-100 dataset; and Figures 37 and 38, respectively, using the non-IID partitioned Shakespeare dataset. Similar conclusions can be made and hence we omit the discussions here.

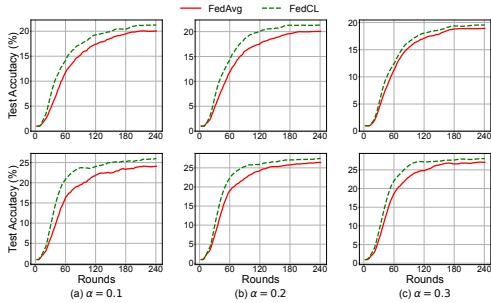

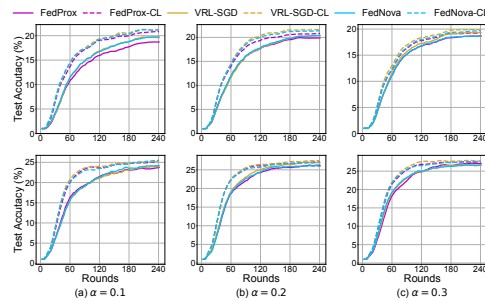

Figure 29: Test accuracy: **(top)** AlexNet and **(bottom)** VGG-11 on non-IID **CIFAR-100**.

Figure 30: Test accuracy: **(top)** AlexNet and **(bottom)** VGG-11 on non-IID **CIFAR-100**.

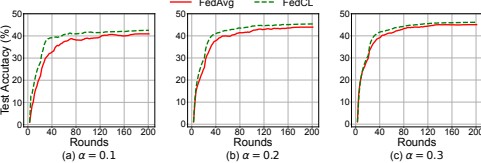

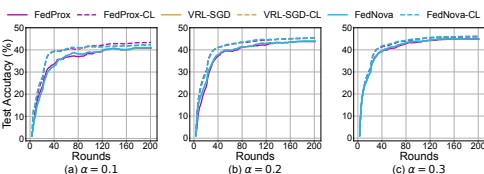

Figure 31: Test accuracy: **LSTM** on non-IID **Shakespeare**.

Figure 32: Test accuracy: **LSTM** on non-IID **Shakespeare**.

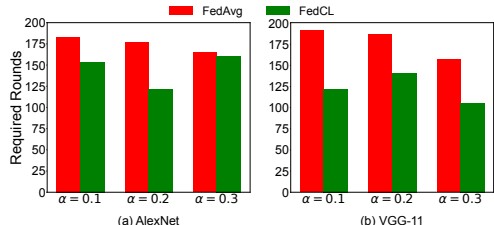

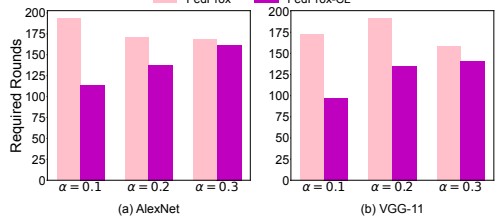

Figure 33: Communication of **FedAvg** and **FedCL** on non-IID **CIFAR-100**.

Figure 34: Communication of **FedProx** and **FedProx-CL** on non-IID **CIFAR-100**.

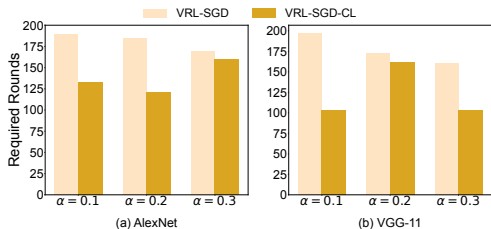

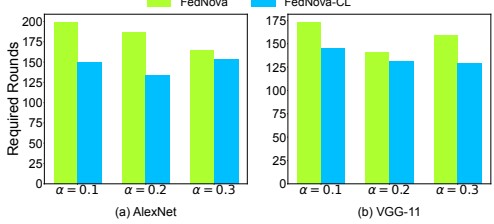

Figure 35: Communication of **VRL-SGD** and **VRL-SGD-CL** on non-IID **CIFAR-100**.

Figure 36: Communication of **FedNova** and **FedNova-CL** on non-IID **CIFAR-100**.

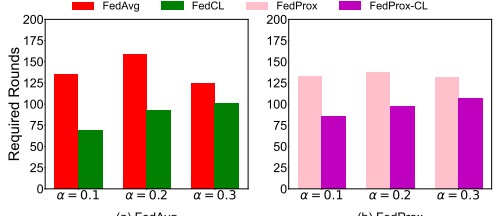

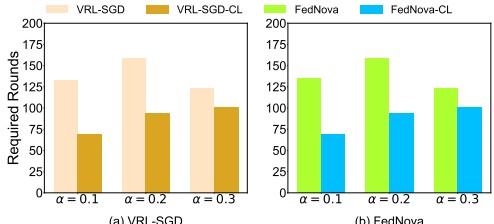

Figure 37: Communication of **FedAvg** vs. **FedCL** and **FedProx** vs. **FedProx-CL** on non-IID **Shakespeare**.

Figure 38: Communication of **VRL-SGD** vs. **VRL-SGD-CL** and **FedNova** vs. **FedNova-CL** on non-IID **Shakespeare**.

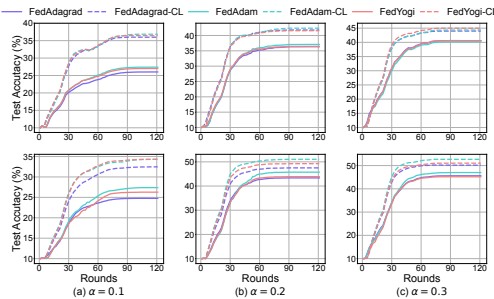 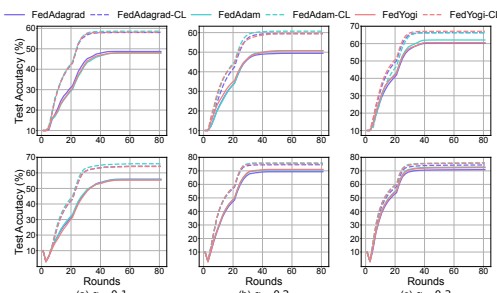

Figure 39: Test accuracy of **FedAdagrad, FedYogi, FedAdam** and **FedAdagrad-CL, FedYogi-CL, FedAdam-CL** using **(top) AlexNet** and **(bottom) VGG-11** on non-IID **CIFAR-10**.

Figure 40: Test accuracy of **FedAdagrad, FedYogi, FedAdam** and **FedAdagrad-CL, FedYogi-CL, FedAdam-CL** using **(top) AlexNet** and **(bottom) VGG-11** on non-IID **Fashion-MNIST**.

### A.3 ADDITIONAL RESULTS ON THE GENERALIZATION OF FEDCL

In Section 5.3, we discuss the generalization of our proposed FedCL by considering three state of the arts, i.e., FedProx (Li et al., 2020a), VRL-SGD (Liang et al., 2019) and FedNova (Wang et al., 2020c). We now provide additional experimental results on the generalization of FedCL by considering FedOPT (Reddi et al., 2021) with three other state-of-the-art methods, i.e., FedAdagrad, FedYogi, and FedAdam. We call the corresponding CL periods-aware methods as FedAdagrad-CL, FedYogi-CL, and FedAdam-CL, respectively. We consider the same setting as that in Section 5.3. From Figures 39 and 40, we again observe that CL periods awareness significantly improves the test performance of baseline methods, i.e., FedAdagrad-CL, FedYogi-CL, and FedAdam-CL outperform FedAdagrad, FedYogi, and FedAdam, respectively, in all scenarios. Likewise, the CL periods augmented method, e.g., FedAdagrad-CL requires fewer rounds to achieve the final test accuracy of the corresponding baseline FedAdagrad, as shown in Figures 41 and 42. Similar observations can be made for FedYogi in Figures 43 and 44, and FedAdam in Figures 45 and 46.

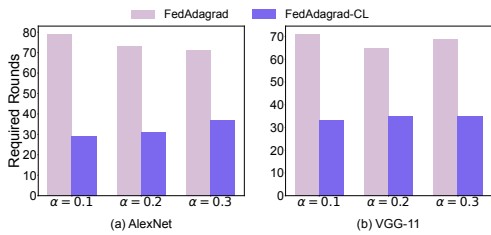 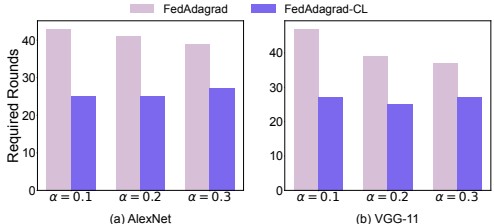

Figure 41: Communication of **FedAdagrad** and **FedAdagrad-CL** on non-IID **CIFAR-10**.

Figure 42: Communication of **FedAdagrad** and **FedAdagrad-CL** on non-IID **Fashion-MNIST**.

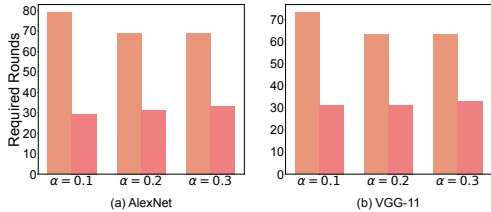 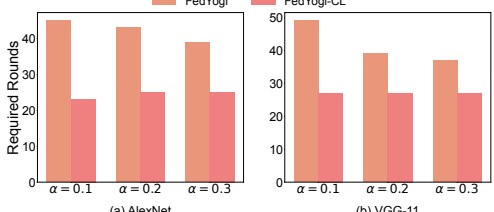

Figure 43: Communication of **FedYogi** and **FedYogi-CL** on non-IID **CIFAR-10**.

Figure 44: Communication of **FedYogi** and **FedYogi-CL** on non-IID **Fashion-MNIST**.

### A.4 RANDOM INCREASE AND DECREASE THE NUMBER OF PARTICIPATED CLIENTS

Besides deterministically increasing or decreasing the number of participated clients as in FedCL, we randomly increase or decrease the number. Specifically, we consider two settings. On the one

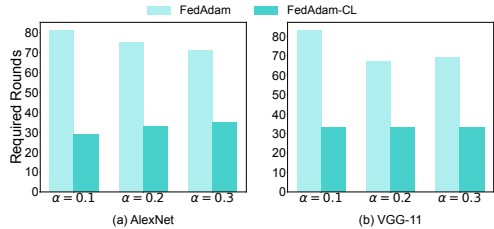

Figure 45: Communication of **FedAdam** and **FedAdam-CL** on non-IID **CIFAR-10**.

Figure 46: Communication of **FedAdam** and **FedAdam-CL** on non-IID **Fashion-MNIST**.

hand, we fix the probability of decreasing the participated clients from m to m/2 to be 30% in each round, and investigate the impact of the probability of increasing the participated clients from m to 2m in each round. On the other hand, we fix the probability of increasing the participated clients from m to 2m to be 30% in each round, and investigate the impact of the probability of decreasing the participated clients from m to m/2 in each round. We report the final test accuracy of the model, and compare it with the FedCL (see Algorithm 2) and the FedAvg. As shown in Figures 47 and 48, random increase or decrease may not necessarily improve the performance of FedAvg. This is due to the fact that the random strategy may not necessarily align with the findings of critical learning periods (e.g., Achille et al. 2019 and Yan et al. 2022) that more data/clients need to be involved in early training phases.

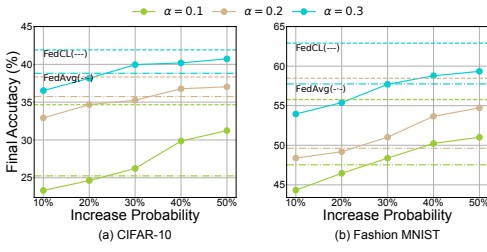
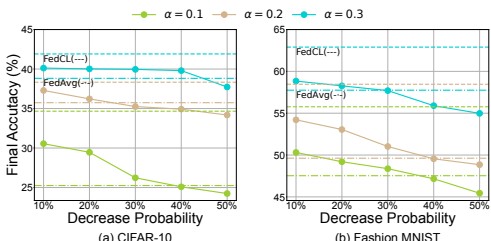

Figure 47: Test accuracy of FedCL when randomly increase the number of participated clients.

Figure 48: Test accuracy of FedCL when randomly decrease the number of participated clients.

## A.5 THE NUMBER OF PARTICIPATED CLIENTS AND THE FGN

We report the number of participated clients and the FGN curve in Figure 49. We observe that more clients are involved in the early training phases where the CL periods occur.

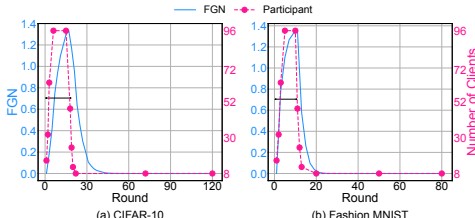

Figure 49: Relationship between FGN and the number of participants.

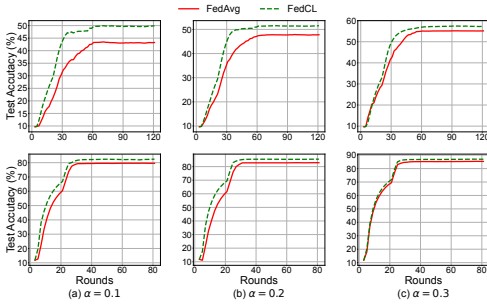

Figure 50: Test accuracy of **FedAvg** and **FedCL** using **ResNet-18** on non-IID **(top)** CIFAR-10 and **(bottom)** Fashion-MNIST.

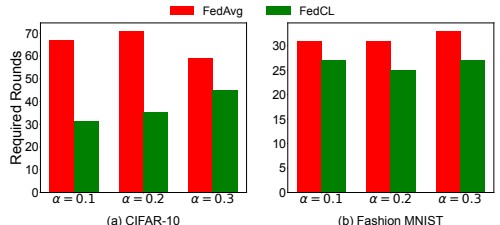

Figure 51: Communication efficiency of **FedAvg** and **FedCL** using **ResNet-18**.

## A.6 RESULTS ON RESNET-18 MODEL

**Importance of CL Periods Awareness: FedCL v.s. FedAvg:** Complementary to Section 5.2, We further evaluate the performance of FedCL using the ResNet-18 model on non-IID partitioned CIFAR-10 and Fashion-MNIST datasets. The test accuracy comparison on non-IID partitioned CIFAR-10 and Fashion-MNIST with FedAvg selecting 16 clients in each round is shown in Figure 50. Again, we observe that FedCL consistently outperforms FedAvg in all scenarios. Similarly, the benefit of CL periods awareness for FL training is further reflected via communication efficiency as shown in Figure 51.

**Generalization:** Complementary to Section 5.3 and Appendix A.3, we study the generalization of FedCL and consider six state of the arts, i.e., FedProx, VRL-SGD, FedNova, FedAdagrad, FedYogi, and FedAdam, respectively, using the ResNet-18 model. The test accuracy comparisons are presented in Figures 52 and 53 with the communication efficiency comparisons presented in Figures 54- 59, respectively. Again, we observe that the CL periods awareness can improve the performance of the state of the arts.

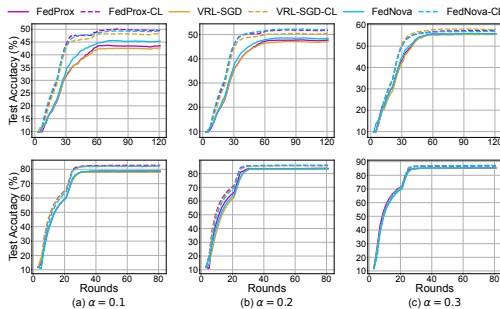

Figure 52: Test accuracy of **FedProx, VRL-SGD, FedNova** and **FedProx-CL, VRL-SGD-CL, FedNova-CL** using **ResNet-18** non-IID **(top)** CIFAR-10 and **(bottom)** Fashion-MNIST.

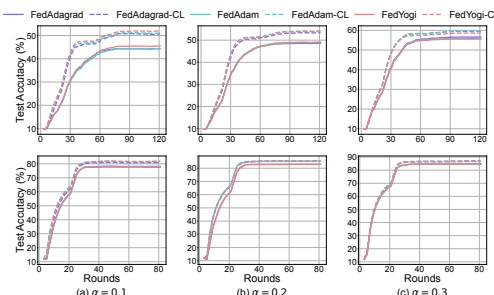

Figure 53: Test accuracy of **FedAdagrad, FedYogi, FedAdam** and **FedAdagrad-CL, FedYogi-CL, FedAdam-CL** using using **ResNet-18** non-IID **(top)** CIFAR-10 and **(bottom)** Fashion-MNIST.

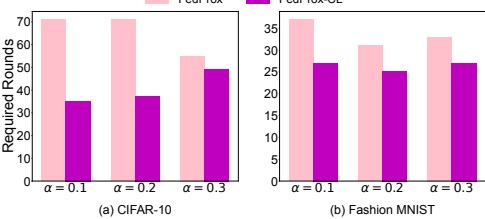

Figure 54: Communication of **FedProx** and **FedProx-CL** using **ResNet-18**.

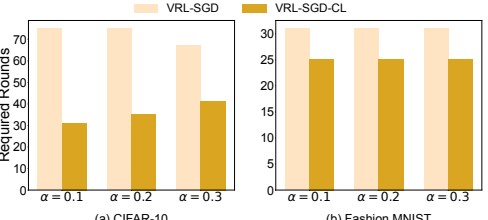

Figure 55: Communication of **VRL-SGD** and **VRL-SGD-CL** using **ResNet-18**.

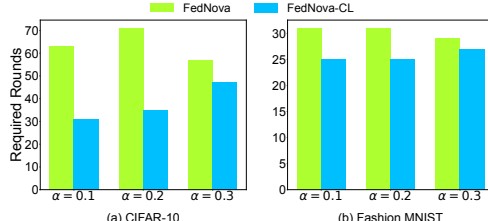
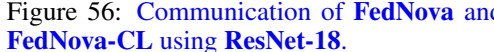

Figure 56: Communication of **FedNova** and **FedNova-CL** using **ResNet-18**.

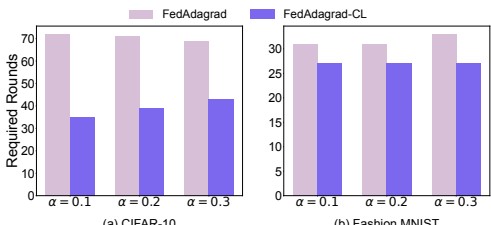

Figure 57: Communication of **FedAdagrad** and **FedAdagrad-CL** using **ResNet-18**.

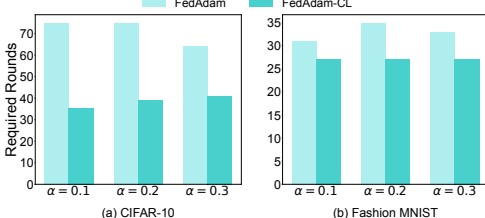

Figure 58: Communication of **FedAdam** and **FedAdam-CL** using **ResNet-18**.

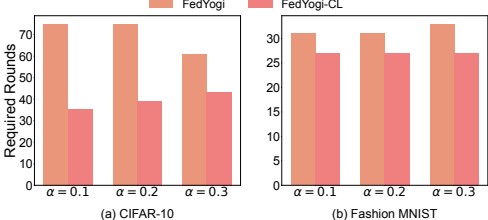

Figure 59: Communication of **FedYogi** and **FedYogi-CL** using **ResNet-18**.

