# OpenReview forum: "FedCL: Critical Learning Periods-aware Adaptive Client Selection in Federated Learning"
_ICLR.cc/2023/Conference — Submitted to ICLR 2023_

### Official Review · Reviewer_dEoL · 2022-10-21

**Confidence:** 4
**Correctness:** 4
**Technical Novelty And Significance:** 2
**Empirical Novelty And Significance:** 2
**Recommendation:** 5

**Clarity, Quality, Novelty And Reproducibility:**

This work is clear and of fair quality. The introduce metric and designed framework are technically solid. What concerns me the most is the novelty and contribution.

**Strength And Weaknesses:**

Strength:
1. the introduced metric to locate the critical learning periods is simple yet effective, which has been proved in the extensive experimental results.
2. the designed FedCL framework and the adaptive client determination strategy can keep a good balance between final performance and communication efficiency.
3. the authors conduct extensive experiments in this work to prove the efficacy and efficiency;

Weakness:
1. the introduced metric that is used to locate the critical learning, is based on local loss changes. However, using local loss values to select active clients in FL is not new. Please refer to the following works.
- Client selection in federated learning: Convergence analysis and power-of- choice selection strategies.arxiv, 2020
- Active federated learning. arrive, 2019
- FedCor: Correlation-Based Active Client Selection Strategy for Heterogeneous Federated Learning, CVPR 2022

2. The strategy of adaptive client selection has been proved in prior works. However, in prior client selection works such as FedCor,  they select the clients based on their data characteristics, rather than simply increasing/decreasing the client numbers. Compared to those prior works, the client selection strategy in this work seems too simple and straightforward.

3. I can not see an experiment using random increasing/decreasing strategy. Did I miss that in some place?

**Summary Of The Paper:**

This work proposes a solution to adaptively set client number on critical learning periods in federated learning. Specifically, the authors introduce a simple yet effective metric to locate the critical learning periods. After locating the critical learning periods, the client number can be adaptively adjusted to keep a good balance between the final performance and communication efficiency.  The introduced metric and the whole framework are proved effective in the extensive experiments.

Contributions:
1. the authors introduced a simple yet effective metric to locate the critical learning periods in federated learning.
2. the authors integrated the introduced metric into different FL methods to construct FedCL framework, which can adaptively determine the number of clients to participate in each training around.

**Summary Of The Review:**

This work proposes a solution to adaptively set client numbers on critical learning periods in federated learning. Specifically, the authors introduce a simple yet effective metric to locate the critical learning periods. After locating the critical learning periods, the client number can be adaptively adjusted to keep a good balance between the final performance and communication efficiency.  The introduced metric and the whole framework are proved effective in the extensive experiments.

What concerns me the most is the novelty and contribution. The introduced metric that is used to locate the critical learning, is based on local loss changes. However, using local loss values to select active clients in FL is not new. I have listed some references in the comments. Please make a comparison with them and list the difference. Other than that, the strategy of adaptive client selection proposed in this work, seems too simple (i.e., increasing/decreasing the active client numbers), compared to the prior works.

I hope the authors can provide more discussions and comparisons in the rebuttal. I am willing to adjust my score if I receive convincing feedback.

---

> ### Author Response · Authors · 2022-11-13
> **Author Response to Reviewer dEoL**
>
> Thank you very much for your review and constructive comments. Here we would like to address the reviewer's concerns and hope that can help raise the rating of our paper. The detailed responses are as follows:
>
> **Your comment:** “The introduced metric that is used to locate the critical learning, is based on local loss changes. However, using local loss values to select active clients in FL is not new. Please refer to the following works.”
>
> **Our response:** Thank you for your insightful comments and pointing out these works.  Please see our response to the “Common comment 1” above. The first paper you mentioned was cited in our submission.  We further cited these two papers you mentioned in “Section 2 Related Work”.
>
> **Your comment:** “The strategy of adaptive client selection has been proved in prior works. However, in prior client selection works such as FedCor, they select the clients based on their data characteristics, rather than simply increasing/decreasing the client numbers. Compared to those prior works, the client selection strategy in this work seems too simple and straightforward.”
>
> **Our response:** Thank you for your insightful comments. See our response to the “Common comment 2” above. We agree with the reviewer that there are many existing works focusing on selecting clients via addressing the statistical/data heterogeneity, including FedCor, and addressing the system heterogeneity.  We would like to reiterate (also refer to our response to “Common Comment 2”) that most of these algorithms focusing on targeting on the federated optimization itself, e.g., selecting clients based on data characteristics that contributing most to the gradients, or using online learning methods (e.g., the Oort method proposed in Lai et al. 2021).
>
> Instead, in this paper, we provide the first attempt to connect the federated optimization and generalization of deep neural networks that are used for federated training. In other words, we leverage the recently discovered phenomenon of the importance of early training phase into the federated optimization, in particular, focusing on the issue of client selection. As motivated by Achille et al. 2019 and Gang et al. 2022 that are among the first to study the connection between centralized/decentralization optimization and generalization via identifying critical learning periods, i.e., if not enough data are used for training in the early phases, then no matter how many additional training are performed in the later stage, the model performance is permanently damaged, we proposed FedCL via increasing the number of selected clients during the CL periods.  Further taking the communication in consideration, FedCL decreased the number of selected clients after the CL periods. We thank again for the reviewer for this insightful comment. Further improving the performance of FedCL, e.g., leveraging ideas in FedCor is an interesting venue to explore in the future work.

---

> > ### Author Response · Authors · 2022-11-13
> > **Author Response to Reviewer dEoL**
> >
> > **Your comment:** “I cannot see an experiment using random increasing/decreasing strategy. Did I miss that in some place?.”
> >
> > **Our response:** Thank you for your insightful comments. As we response above in “Response to Common Response 1”, in this paper, we are among the first to connect the federated optimization and the generalization of deep neural network training via the critical learning periods. As motivated by Achille et al. 2019 and Yan et al. 2022, if not enough data are leveraged for training in the early phases, the final model accuracy will be permanently degraded no matter how many additional trainings are performed in the later stage.  To this end, in our framework, we select as many clients as possible in the early training phases, i.e., we quickly increase the number of clients (by doubling the number in each round).  If we always select the largest number of clients, the comparison with FedAvg may be unfair, and so we decrease the number of clients after the critical learning periods.  To this end, even though we select a larger number of clients in the early phases, the average number of participated clients in each around is close to that of FedAvg.
> >
> > We thank the reviewer for pointing out this issue and suggestions.  Herein, we provide additional experimental results.  In other words, instead of deterministically increase or decrease the number of participated clients, we randomly increase or decrease the number per the reviewer’s suggestions.  Specifically, we consider two settings.  On the one hand, we fix the probability of decreasing the participated clients from m to m/2 to be 30% in each round, and investigate the impact of the probability of increasing the participated clients from m to 2m in each round.  On the other hand, we fix the probability of increasing the participated clients from m to 2m to be 30% in each round, and investigate the impact of the probability of decreasing the participated clients from m to m/2 in each round.  We report the final test accuracy of the model, and compare it with the FedCL in this paper and the FedAvg.  As shown in Figures 47 and 48 in Appendix A.4, random increase or decrease may not necessarily improve the performance of FedAvg.  This is due to the fact that the random strategy may not necessarily align with the findings of critical learning periods (e.g., Achille et al. 2019 and Yan et al. 2022) that more data/clients need to be involved in early training phases.

---

> > > ### Author Response · Authors · 2022-11-16
> > > **Follow-up with Reviewer dEoL**
> > >
> > > Since the reviewer-author discussion period is ending soon, we just wanted to check in and ask if our rebuttal clarified and answered your questions. We would be very happy to engage further if there are additional questions.
> > >
> > > Also, we wanted to check if our additional clarifications regarding the merits of the paper would convince the reviewer to raise the score. Thank you!

---

### Official Review · Reviewer_phLC · 2022-10-23

**Confidence:** 4
**Correctness:** 4
**Technical Novelty And Significance:** 2
**Empirical Novelty And Significance:** 2
**Recommendation:** 6

**Clarity, Quality, Novelty And Reproducibility:**

The paper is clear in its motivation and objectives. The idea is simple and original.  The authors provide code for reproducibility.

**Strength And Weaknesses:**

## Strengths
- The idea to leverage gradient norms to detect critical learning periods is good.
- Results for three different models show significant performance improvements.
- The algorithm is very simple to implement and can be used on top of most state-of-the-art FL algorithms
## Weaknesses
- The datasets used `CIFAR-10` and `Fashion MNIST` is fairly simple. It would have been more convincing to see the results on more complex data like `CIFAR-100`. Additionally, the models used are too simple. It would be more convincing to show results on `Resnet-18` and `Mobile Net` if possible [1].
- The baseline accuracy results using Federated Fisher Information are not given.
- It is unclear how the federated gradient norm predicts critical learning periods.
- It would be better to show the value of the federated gradient norm and the number of clients being sampled alongside the learning curve to better understand when critical learning periods occur.
- It is not clear when the algorithm predicts a critical learning period and if it is effective. From my understanding, it should have some dependency on the random seed used to select clients randomly. Experiments should be run on multiple seeds to check if the algorithm finds critical learning periods at different points in training.
- The authors claim FedAvg is state-of-the-art which may not be true. Results on new algorithms like FedOPT[2] should be shown as well.
- A major question is when the critical learning period occurs. Does it always occur in the early optimization stage? If so, is detecting CL necessary? I think then a simple learning rate scheduling algorithm could work.

[1] Wang, Jianyu, et al. "A field guide to federated optimization." arXiv preprint arXiv:2107.06917 (2021).

[2] Reddi, Sashank, et al. "Adaptive federated optimization." arXiv preprint arXiv:2003.00295 (2020).



**Summary Of The Paper:**

The paper proposes a novel approach to efficiently identify the critical learning (CL) periods and designs a CL period-aware federated learning (FL) algorithm. The algorithm adjusts the number of selected clients by increasing the number of clients per round during CL periods and returning gradually to normal when outside those periods. The authors claim that the algorithm augments state-of-the-art FL algorithms by a considerable margin and shows extensive empirical results to justify their approach.

**Summary Of The Review:**

Although the paper proposes a simple but original idea, its evaluation is made difficult as the experiments and ablation studies are not convincing enough. The experiment section needs to be improved if not revamped and more theoretical foundations for critical learning periods and their detection should be added if possible.

---

> ### Author Response · Authors · 2022-11-13
> **Author Response to Reviewer phLC**
>
> Thank you very much for your review and constructive comments. Here we would like to address the reviewer's concerns and hope that can help raise the rating of our paper. The detailed responses are as follows:
>
> **Your comment:** “The datasets used CIFAR-10 and Fashion MNIST is fairly simple. It would have been more convincing to see the results on more complex data like CIFAR-100. Additionally, the models used are too simple. It would be more convincing to show results on Resnet-18 and Mobile Net if possible [1].”
>
> **Our response:** First, we actually have results on CIFAR-100.  Specifically, in this paper, we considered two tasks: image classification using CIFAR-10, Fashion MNIST and CIFAR-100, and natural language processing task on next-character prediction using the dataset of The Complete Works of William Shakespeare (Shakespeare). Due to space constraints and for ease of discussions, as we mentioned at the beginning of Section 5, “We relegate details of datasets and models, and additional experimental results, particularly on CIFAR-100 and Shakespeare, to Appendix A”.  In particular, the results on CIFAR-100 are provided in Figures 29, 30, 33, 34, 35, 36, and on Shakespeare in Figures 31, 32, 37, 38 with discussions available in Appendix A.2.3. Therefore, we not only use the CIFAR-10 and Fashion MNIST datasets, but also more complex CIFAR-100 and Shakespeare datasets.
>
> Second, we consider two machine learning models AlexNet and VGG-11, which are widely used in the literature. We agree that AlexNet is simple, but VGG-11 is not that much simple as shown in Table 1. Furthermore, many state-of-the-art works, e.g., Yang et al. 2021, Cho et al. 2021, Ruan et al. 2021, Wang et al. 2020c, Haddadpour et al. 2021, Wang et al. 2020a, etc (All were cited in the paper) mainly used relatively simple models, such as LR (logistic regression), 2NN, standard CNN with CIFAR-10, MNIST, Fashion MNIST, EMNIST datasets, while we consider not only the image classification tasks (also use CIFAR-100 dataset), but also consider the next-character prediction task (using the Shakespeare dataset). Though we believe that the extensive experimental results using different models and datasets under various settings (including an ablation study in Section 5.4) in the paper can support our key ideas and conclusions, we agree with the reviewer that using more models will add extra values to the paper. We are running additional results using the RestNet-18 model, and we will provide the results in the coming days, which we believe that the same observations and conclusions can be made.
>
> [Yang et al. 2021] Haibo Yang, Minghong Fang, and Jia Liu. Achieving Linear Speedup with Partial Worker Participation in Non-IID Federated Learning. In Proc. of ICLR, 2021.
>
> [Cho et al. 2022] Yae Jee Cho, Jianyu Wang, and Gauri Joshi. Towards understanding biased client selection in federated learning. In Proc. of AISTATS, 2022.
>
> [Ruan et al. 2021] Yichen Ruan, Xiaoxi Zhang, Shu-Che Liang, and Carlee Joe-Wong. Towards Flexible Device Participation in Federated Learning. In Proc. of AISTATS, 2021.
>
> [Wang et al. 2020c] Jianyu Wang, Qinghua Liu, Hao Liang, Gauri Joshi, and H Vincent Poor. Tackling the Objective Inconsistency Problem in Heterogeneous Federated Optimization. Proc. of NeurIPS, 2020c.
>
> [Haddadpour et al. 2021] Farzin Haddadpour, Mohammad Mahdi Kamani, Aryan Mokhtari, and Mehrdad Mahdavi. Federated Learning with Compression: Unified Analysis and Sharp Guarantees. In Proc. of AISTATS, 2021.
>
> [Wang et al. 2020a] Hao Wang, Zakhary Kaplan, Di Niu, and Baochun Li. Optimizing Federated Learning on Non-IID Data With Reinforcement Learning. In Proc. of IEEE INFOCOM, 2020a.
>
> **Your comment:** “The baseline accuracy results using Federated Fisher Information are not given.”
>
> **Our response:** See our response to the “Common comment 2” above.
>
> **Your comment:** “It is unclear how the federated gradient norm predicts critical learning periods. It would be better to show the value of the federated gradient norm and the number of clients being sampled alongside the learning curve to better understand when critical learning periods occur.”
>
> **Our response:** Thank you for your insightful comments and suggestions. We plot the number of participated clients alongside the FGN and show it in Figures 49 in Appendix A.5. Again, as observed that more clients are involved in the early training phases, which is consistently with the observations made in Achille et al. 2019 and Yan et al. 2022.  Also refer to our response to the Common Comment 1 above.

---

> > ### Author Response · Authors · 2022-11-13
> > **Author Response to Reviewer phLC**
> >
> > **Your comment:** “It is not clear when the algorithm predicts a critical learning period and if it is effective. From my understanding, it should have some dependency on the random seed used to select clients randomly. Experiments should be run on multiple seeds to check if the algorithm finds critical learning periods at different points in training.”
> >
> > **Our response:** Thank you for this insightful comment and provide us an opportunity to provide additional experiment results to clarify this issue. We further evaluate the robustness of our proposed FGN on identifying the CL periods, which further improve the model accuracy when leveraged in FedCL. Specifically, we randomly generate five seeds and report the CL period identified in Figure 18, and test accuracy in Figure 19, with discussions provided in Appendix A.2.1.  It is clear that our FGN can consistently and robustly identify the CL periods, which are almost the same across different seeds.
> >
> > **Your comment:** “The authors claim FedAvg is state-of-the-art which may not be true. Results on new algorithms like FedOPT[2] should be shown as well.”
> >
> > **Our response:** Thank you for this insightful comment.  As we discussed in Section 3, FedAvg is the first and the most widely used one (the FedOPT also discussed and cited FedAvg). Hence in this paper, we see FedAvg as our basic block to illustrate the ideas of augmenting client selection with CL periods. Our proposed CL periods-aware FL framework FedCL, HOWEVER, is orthogonal to and can be easily combined with most existing state-of-the-art methods.  In other words, FedCL can be easily combined with other methods by simply replacing the inner optimization subroutine (FedAvg) in Algorithm 2 with the corresponding state of the arts.  In particular, we evaluate the generalization of our FedCL framework by considering three state-of-the-art methods FedProx, VRL-SGD and FedNova in Section 5.3 with additional results provided in the appendix. These generalization results strongly support the importance of being CL periods awareness and the generalization of our proposed FedCL framework.
> >
> > We thank the reviewer for pointing out the state-of-the-art FedOPT. Similar to Section 5.3 for evaluating FedProx, VRL-SGD and FeNova, we further evaluate the generalization of FedCL when combing with the three methods proposed in FedOPT, i.e., FedAdagrad, FedYogi, and FedAdam. We call the corresponding CL periods-aware methods as FedAdagrad-CL, FedYogi-CL, and FedAdam-CL, respectively. We report the experimental results in Appendix A.3. From Figures 39 and40, we again observe that CL periods awareness significantly improves the test performance of baseline methods, i.e., FedAdagrad-CL, FedYogi-CL, and FedAdam-CL outperform FedAdagrad, FedYogi, and FedAdam, respectively, in all scenarios. Likewise, the CL periods augmented method, e.g., FedAdagrad-CL requires fewer rounds to achieve the final test accuracy of the corresponding baseline FedAdagrad, as shown in Figures 41 and 42. Similar observations can be made for FedYogi in 43 and 44, and FedAdam in Figures 45 and 46.
> >
> > For ease of presentation and the rebuttal purpose, we currently add these results in Appendix A.3.  Per the reviewer’s approval, we will merge these results with those in Section 5.3.  For instance, we will discuss FedOPT in Section 5.3, and merge Figures 6 and 39, and Figures 7 and 40, to illustrate the generalization of FedCL by considering these six state-of-the-art methods.
> >
> > **Your comment:** “A major question is when the critical learning period occurs. Does it always occur in the early optimization stage? If so, is detecting CL necessary? I think then a simple learning rate scheduling algorithm could work.”
> >
> > **Our response:** Thank you for this insightful comments. We provide an answer to the generalization issue in “Common Comment 1”.  In addition, we would like to mention that   Achille et al., 2019 and Yan et al., 2022 demonstrated the existence of CLP when using fixed learning rate and scheduling learning rate, and indicated that enough training samples during CLP are necessary to improve performance. Though using a fine-tuned learning rate scheduling algorithm, the CLP-based method could still improve its performance. To this end, in our framework, we select as many clients as possible in the early training phases, i.e., we quickly increase the number of clients (by doubling the number in each round).  If we always select the largest number of clients, the comparison with FedAvg may be unfair, and so we decrease the number of clients after the critical learning periods.  To this end, even though we select a larger number of clients in the early phases, the average number of participated clients in each around is close to that of FedAvg.

---

> > > ### Author Response · Authors · 2022-11-13
> > > **Author Response to Reviewer phLC**
> > >
> > > **Your comment:** “[3] Nguyen, Luong Trung, Junhan Kim, and Byonghyo Shim. "Gradual federated learning with simulated annealing." IEEE Transactions on Signal Processing 69 (2021): 6299-6313.”
> > >
> > > **Our response:** In your review, you mentioned the above paper, however, we did not see any comment in your review related to [3].

---

> > > > ### Author Response · Authors · 2022-11-15
> > > > **Author Response to Reviewer phLC**
> > > >
> > > > **Your comment:** “The datasets used CIFAR-10 and Fashion MNIST is fairly simple. It would have been more convincing to see the results on more complex data like CIFAR-100. Additionally, the models used are too simple. It would be more convincing to show results on Resnet-18 and Mobile Net if possible [1].”
> > > >
> > > > **Our response:**  Following our previous response, we now provide additional results using the ResNet-18 model.  Complementary to Section 5.2, we first compare FedCL and FedAvg using ResNet-18 on non-IID partitioned CIFAR-10 and Fashion-MNIST to illustrate the importance of CL periods awareness. The results are presented in Figures 50 and 51 in Appendix A.6.  Then complementary to Section 5.3, we evaluate the generalization of FedCL by considering six state of the arts, i.e., FedProx, VRL-SGD, FedNova, and three methods in FedOPT (i.e., FedAdagrad, FedAdam, FedYogi) as suggested by the reviewer.  The results are presented in Figures 52-59 in Appendix A.6. From those results, we again reach the same conclusions as we have in the paper.
> > > >
> > > > As no paper is able to run all existing models using all existing datasets, neither does this paper.  However, we believe that our extensive experiments on two tasks (image classification, and next-character prediction) using CIFAR-10, CIFAR-100, Fashion-MNIST and the Shakespeare datasets, with representative DNN models including AlexNet, VGG-11 and ResNet-18, along with a comprehensive ablation study (Section 5.4) strongly and consistently support the major contributions in this paper.  Due to time constraints on rebuttal and revisions, we will run other results using MobileNet etc, and include these results in the final version.
> > > >
> > > > Note that for the purpose of rebuttal, we now position these results in Appendix A.6. Per the reviewer’s approval, we will combine these results on ResNet-18 with those on AlexNet and VGG-11 in our final version.

---

> > > ### Comment · Reviewer_phLC · 2022-11-16
> > > **Response to Authors**
> > >
> > > Thanks for doing the additional experiments. I have improved my score.  Please also consider adding a line in the experiment section explaining the reasoning behind your model and dataset selection and what previous works were followed and why they would be good benchmarks for this task in particular. Please consider merging FedOPT results on the main paper if possible considering page limits.

---

> > > > ### Author Response · Authors · 2022-11-16
> > > > **Thank you!**
> > > >
> > > > Thank you very much for improving the score.  We appreciate it.  In our camera-ready version, we will add the explanation in the experiment section as you suggested.  We will also merge FedOPT results with those in Section 5.3 (e.g., merge Figures 6 and 39, and Figures 7 and 40) to illustrate the generalization of FedCL by considering all these six state-of-the-art methods.

---

### Official Review · Reviewer_8m8J · 2022-10-23

**Confidence:** 4
**Correctness:** 3
**Technical Novelty And Significance:** 2
**Empirical Novelty And Significance:** 4
**Recommendation:** 5

**Clarity, Quality, Novelty And Reproducibility:**

The paper is mostly well-written with sufficient discussions about its settings and implementation details. However, I do have several questions need authors’ clarification.

- It is unclear how the sampling based on Dirichlet distribution was performed to create heterogeneity among clients. It would be great if the authors could provide more details on this setup in the discussion phase.
- I am a bit confused about the $\Delta l_k(\cdot)$ used in Eq (1): is it the actual change of training loss or the approximated one using the gradient?

The novelty of this work is relatively speaking limited, it is largely based on a previous work’s finding about the existence of critical learning period in federated learning and simplify the previous metric using the change of loss function. All the provided results are empirical, and it is hard to justify if the proposed solution can completely solve the problem.


**Strength And Weaknesses:**

Strength:
+ The proposed new metric is very straightforward and easy to implement, which ensures its applicability in practice.
+ The authors provided very extensive experimentation about the proposed solution under various settings, e.g., different degree of heterogeneity across clients, different federated learning algorithms, and various settings of its hyper-parameters. This provides a comprehensive collection of evidence about the effectiveness of the proposed solution.
+ The manuscript is well-written and easy to follow.

Weakness:
- As the cause of critical learning period is unclear, it is hard to tell if the proposed solution can fundamentally solve the problem. Throughout the paper, including the related work section, there is no discussion about why the critical learning period is happening. Is it due to non-convex optimization? And why simply increasing the number of training instances will solve the problem? If it is due to non-convexity, the starting point also matters. The performed experiments seem to suggest heterogeneity is the main cause of critical learning period. But it is never explicitly articulated. Without any analysis about the mechanism of critical learning period, it is hard to justify why the proposed solution could solve the problem.
- The paper argues that permanent damage in model training can be made when the critical learning period is not well handled, no matter how many more training instances could be provided later on. But as the proposed solution could not address the issue either: if it detects the current round is in the critical learning period, it only increases the number of clients for the next round; but the damage has already been made in this round. In addition, client sampling can also cause variance in FGN calculation, which can misclassify the critical learning period. A false negative detection would introduce the damage. And hence, it is still unclear why the proposed solution could be helpful.

Questions:
- Even under the proposed solution, the clients are still randomly sampled. Why do not we select the clients who can best help us circumvent the critical learning period? For example, find a set of clients whose local FGN is smallest?
- Although FedFIM is expensive to compute, it is important to compare how effective its identified critical learning period is for improving federated learning comparing to FGN’s.
- Since the communication cost is defined by the total number of communication rounds, instead of communicated bits, why do not we start with all clients? And how would the hyper-parameter m affect the solution’s performance?
- How would the solution perform if the environment is homogeneous? For example, when we uniformly distribute the dataset onto clients, what’s the performance gain against FedAvg?


**Summary Of The Paper:**

This paper leverages the concept of critical learning period for client selection in federated learning, with the purpose of improving communication efficiency while maintaining or even improving the learning effectiveness. Built based on a previous work that identified the existence of critical learning period in federated learning, this work proposes a light-weighted metric to detect the critical learning period in each round of federated model update, and adaptively change the number of selected clients for gradient update. Extensive experiment results on two benchmark datasets, several base models and federated learning algorithms demonstrated the promising performance gain of the proposed solution.

**Summary Of The Review:**

The provided solution is simple and intuitive, and its effectiveness is proved via an extensive set of experiments. However, this paper lacks necessary investigation of why the critical learning period happens (e.g., what’s the root cause), and therefore it is hard to justify whether proposed solution can completely address the problem.

---

> ### Author Response · Authors · 2022-11-13
> **Author Response to Reviewer 8m8J**
>
> Thank you very much for your review and constructive comments. Here we would like to address the reviewer's concerns and hope that can help raise the rating of our paper. The detailed responses are as follows:
>
> **Your comment:** “Even under the proposed solution, the clients are still randomly sampled. Why do not we select the clients who can best help us circumvent the critical learning period? For example, find a set of clients whose local FGN is smallest?”
>
> **Our response:** Achille et al. 2019 and Yan et al. 2022 revealed the existence of CL periods in deep neural network training, which may arise from information processing. The information in the weights does not necessarily increase monotonically during the training process. Instead, a rapid growth in information (“memorization phase”) is followed by a reduction of information (“forgetting” phase), even as the classification performance keeps increasing. This behavior is consistent across different tasks and network architectures as shown in Achille et al. 2019, Yan et al. 2022 and among others. Critical learning periods are centered in the memorization phase.
>
> **Your comment:** “Although FedFIM is expensive to compute, it is important to compare how effective its identified critical learning period is for improving federated learning comparing to FGN’s.”
>
> **Our response:** See our response to the “Common comment 2” above.
>
> **Your comment:** “Since the communication cost is defined by the total number of communication rounds, instead of communicated bits, why do not we start with all clients? And how would the hyper-parameter m affect the solution’s performance?”
>
> **Our response:** Thank you for your insightful comments and suggestions.  First, we indeed conducted an ablation study to study the impact of the hyper-parameter $m$ in Section 5.4 with the title of “Client Participation Rate”. Specifically, in Figure 15, we vary the participation rate from 9.3% to 31.2%, which corresponds to the value of $m$ from 12 to 40. As shown in Figure 15, when a state-of-the-art method is augmented with the CL periods, the final test accuracy is consistently improved across all client participation rates. The advantage is particularly pronounced with a low participation rate. This is quite intuitive since in our CL periods aware framework, FedCL selects more clients during the CL periods than FedAvg, and hence the benefits are more obvious when FedAvg has a low client participation rate. We select 16 clients, i.e., a 12.5% participation rate for all state-of-the-art methods. Note that in practice, there is often a low participation rate in FL systems, which are also the settings widely considered in the literature, e.g., Nishio et al. 2019, Karimireddy et al. 2020, Wang et al. 2020c, Yan et al. 2022, Yang et al. 2021.
>
> Second, it is possible to start with all clients from the very beginning. As we response above in “Common Comment 1”, in this paper, we are among the first to connect the federated optimization and the generalization of deep neural network training via the CL periods. As motivated by Achille et al. 2019 and Yan et al. 2022, if not enough data are leveraged for training in the early phases, the final model accuracy will be permanently degraded no matter how many additional trainings are performed in the later stage. To this end, in our framework, we select as many clients as possible in the early training phases, i.e., we quickly increase the number of clients (by doubling the number in each round).  If we always select the largest number of clients, the comparison with FedAvg may be unfair, and so we decrease the number of clients after the CL periods.  To this end, even though we select a larger number of clients in the early phases, the average number of participated clients in each around is close to that of FedAvg.

---

> > ### Author Response · Authors · 2022-11-13
> > **Author Response to Reviewer 8m8J**
> >
> > **Your comment:** “How would the solution perform if the environment is homogeneous? For example, when we uniformly distribute the dataset onto clients, what’s the performance gain against FedAvg?”
> >
> > **Our response:** Thank you for your insightful comments. We indeed experimentally investigated the impact of “Non-IID Degree” in our ablation study in Section 5.4. Need to mention that we simulate the heterogeneous setting by using the Dirichlet distribution, a technique that has been widely used in the literature, e.g., Wang et al. 2020b, Wang et al. 2020c, Lin et al. 2020, Achituve et al 2021, etc (All were cited in the paper). The level of heterogeneity among local datasets across different clients can be reduced when the parameter $\alpha$ increases, e.g., when $\alpha$ get closes to 1, the setting is almost homogeneous. As reported in Figure 11, the CL periods awareness consistently improves the final test accuracy of a state-of-the-art method across all values of $\alpha$ in consideration, where $\alpha=0.9$ leads to an almost homogeneous setting.
> >
> > [Wang et al. 2020b] Hongyi Wang, Mikhail Yurochkin, Yuekai Sun, Dimitris Papailiopoulos, and Yasaman Khazaeni. Federated Learning with Matched Averaging. In Proc. of ICLR, 2020b.
> >
> > [Wang et al. 2020c] Jianyu Wang, Qinghua Liu, Hao Liang, Gauri Joshi, and H Vincent Poor. Tackling the Objective Inconsistency Problem in Heterogeneous Federated Optimization. Proc. of NeurIPS, 2020c.
> >
> > [Lin et al. 2020] Tao Lin, Lingjing Kong, Sebastian U Stich, and Martin Jaggi. Ensemble distillation for robust model fusion in federated learning. Proc. of NeurIPS, 2020.
> >
> > [Achituve et al. 2021] Idan Achituve, Aviv Shamsian, Aviv Navon, Gal Chechik, and Ethan Fetaya. Personalized federated learning with gaussian processes. Proc. of NeurIPS, 2021.
> >
> > **Your comment:** “It is unclear how the sampling based on Dirichlet distribution was performed to create heterogeneity among clients. It would be great if the authors could provide more details on this setup in the discussion phase.”
> >
> > **Our response:** We thank the reviewer to point out this issue and provide us an opportunity to clarify it. The way of simulating a heterogeneous setting by using the Dirichlet distribution is a technique that has been widely used in the literature, e.g., Wang et al. 2020b, Wang et al. 2020c, Lin et al. 2020, Achituve et al 2021, etc (All were cited in the paper). Specifically, for each class of samples, set the class probability in each client by sampling from a Dirichlet distribution with the same $\alpha$ parameter. For instance, when $\alpha=0.5$, sampling $p_k \sim\text{Dir}(0.5)$ and then allocating a $p_{k,j}$ proportion of the training instances of class $k$ to local client $j$. We provide these additional discussions in Appendix A.1.
> >
> > **Your comment:** “I am a bit confused about the Δlk(⋅)  used in Eq (1): is it the actual change of training loss or the approximated one using the gradient?.”
> >
> > **Our response:** It is the approximated one using the gradient. As discussed above (1), the training loss can be approximated by the gradient norm using Taylor expansion, and hence we call it the FGN in (1).

---

> > > ### Author Response · Authors · 2022-11-16
> > > **Follow-up with Reviewer 8m8J**
> > >
> > > Since the reviewer-author discussion period is ending soon, we just wanted to check in and ask if our rebuttal clarified and answered your questions. We would be very happy to engage further if there are additional questions.
> > >
> > > Also, we wanted to check if our additional clarifications regarding the merits of the paper would convince the reviewer to raise the score. Thank you!

---

> > > ### Comment · Reviewer_8m8J · 2022-11-23
> > > **Thanks for the detailed responses**
> > >
> > > I highly appreciated the authors' detailed comments and the additional empirical results.
> > >
> > > As explained in my original review, my main concern regarding this paper is still the vaguely defined concept "critical learning period", especially the cause of it. If we do not know why it is happening, it is very hard to believe one can solve it, or its related problems. For example, the newly provided explanation based on "information processing" does not explain why it is happening, but provides another way to describe the phenomenon.
> > >
> > > I understand the authors' argument that identifying the cause of critical learning period is beyond the scope of this paper; but an opaque foundation might lead to a superficial solution.

---

> > > > ### Author Response · Authors · 2022-11-23
> > > > **Author Response to Reviewer 8m8J**
> > > >
> > > > Thank you very much again for your insightful comments.  We would like to clarify two statements: (i) “Critical learning periods” (CL periods) have been revealed by state of the arts to exist in the training phases of DNNs, both in centralized settings and federated learning settings; and (ii) This paper is one of the first to leverage this phenomenon to improving the algorithm design (in particular for client selection) in federated learning settings. This is motivated by the fact that most existing works only focused on the federated optimization itself (to address different challenges, e.g., statistical and system heterogeneity), and largely ignored the impact of DNNs that are used to training the federated optimization algorithms. In the following, we further elaborate on these two statements.
> > > >
> > > > First, we did not claim that this paper revealed the existing of CL periods in DNNs. As stated earlier, Achille et al. 2019 is among the first to reveal the existence of CL periods in centralized DNN training and Yan et al. 2022 is among the first to reveal this phenomenon in federated settings. In Achille et al. 2019, critical periods are defined to be the time windows of early post-natal development during which sensory deficits can lead to permanent skill impairment, as motivated by studies in humans and animals. Achille et al. 2019 revealed that such phenomenon also exists in the training phases of DNN, in other words, **”DNNs respond to sensory deficits in ways similar to those observed in humans and animals models, which suggested that critical periods may arise from information processing”**.  In particular, Achille et al. 2019 tried to explain this phenomenon vis using the Fisher Information of the weights to measure the effective connectively between layers of a network during training.  Achille et al. 2019 revealed that the information rises rapidly in the early phases of training, and then decreases.  Their analysis suggested that the **first few (initial) training phases are critical in determining the outcome of the training process**. Yan et al. generalized this result to federated settings via Federated Fisher Information (FedFIM).
> > > >
> > > > In short, Achille et al. 2019 and Yan et al. 2022 revealed the existing of CL periods in DNNs via extensive experiments across different tasks and network architectures. Hence, this phenomenon is believed by the community and hence we do not need to solve it. The question is how to efficiently detect or identify the CL periods during the training process (and then how to leverage it into the algorithm design, see below). Different from FedFIM used by Yan et al, we proposed a lightweight FGN to detect CL periods. We agree with the reviewer that a fundamental understanding of CL periods in DNNs, both under centralized and federated settings, will bring benefit to the community.  This is an open problem.
> > > >
> > > > Second, the CL periods have already been leveraged to study different problems in centralized training, e.g., generalization (Jastrzebski et al. 2021), model weight impacts (Frankle et al. 2020), among others (e.g., Golatkar et al. 2019, Jastrzebski et al. 2019, Jastrzebski et al. 2020). Note that all these findings are based on extensive experiments. We are among the first to leverage it into the federated algorithm design. Our extensive experimental studies strongly support our claims that by augmenting existing FL algorithms with the existing of CL periods, the corresponding performance can be significantly improved.  As stated in Section 4.2, we hypothesize that the SGD is navigating to the steeper parts of the loss surface of the global model during CL periods since a larger amount of data samples have contributed to the global model. Since the existing of CL periods in DNNs have been widely verified, the study of this work is not superficial as we defined new metric to detect CL periods, and new framework FedCL to leverage it into the federated algorithm design.
> > > >
> > > > This also leads to some open problems, such as is it possible to directly augment the federated learning optimization (e.g., the objective function) with the CL periods? If possible, how to design the CL periods-aware objective function (i.e., instead of just minimizing the loss itself)?
> > > >
> > > > We thank the reviewer again for this insightful comment.  We would like to add these discussions as part of the **“Limitation and Future Work”** Section to be added in the final version of the paper.

---

> > ### Comment · Reviewer_8m8J · 2022-11-23
> > **Communication cost?**
> >
> > Just to clarify: if the communication cost is defined by the number of communication rounds, we can do the same thing to FL baselines, e.g., enable all clients in each round. Why would this be unfair to FedAvg?
> >
> > My original question meant to ask: will it be ideal if we have all clients participate in FL each round, since it will not increase the referred communication cost.

---

> > > ### Author Response · Authors · 2022-11-23
> > > **Author Response to Communication Cost**
> > >
> > > Thank you very much again for this question.  We answer this question from two perspectives.
> > >
> > > First, in FedAvg itself or in most practical federated learning settings, there is often a large number of clients (i.e., M clients) in the system. In each training round, the parameter setter (PS) only selects a subset of clients (i.e., m clients) to participate.  In practice, m is often much smaller than M.  Hence, from a practical perspective, FedAvg only selects a subset of clients, rather than all clients in each round.
> > >
> > > Second, as we stated in Introduction, we build upon Yan et al. 2022 who showed that if the training dataset for each client is not recovered to the entire training dataset early enough in the training process, the test accuracy of FL is permanently impaired.  We extended this notation to client selection in FL and show that a larger number of clients are only required during these CL periods.  As shown in Algorithm 1, FedCL selects more clients than FedAvg **only** during the CL periods (which actually only account for a small percentage of the whole training process, e.g., see Figure 1).
> > >
> > > However, by doing so, the accuracy comparison may be not fair to FedAvg since FedCL selects more clients during the CL periods. Therefore, after the CL periods, FedCL selects fewer clients than FedAvg. This is also related to the communication overhead/cost question.  During the CL periods, FedCL has a larger communication overhead as it selects more clients than FedAvg; while FedCL has a smaller communication overhead than FedAvg after the CL periods. To this end, the average communication overhead of FedCL is almost close to that of FedAvg in our experiments.  For example, FedAvg selects 16 clients in each round, the average number of clients selected by FedCL is 16.46 using the CIFAR-10 dataset.  We thank the reviewer again for this comment, and we would like to add these discussions along with some experimental results to the final version of the paper.
> > >
> > > Note that the communication rounds are also related to the training time.  In other words, FedCL needs fewer rounds (i.e., less training time) to achieve the same test accuracy as that of FedAvg.

---

### Official Review · Reviewer_29oH · 2022-10-24

**Confidence:** 4
**Correctness:** 3
**Technical Novelty And Significance:** 2
**Empirical Novelty And Significance:** 3
**Recommendation:** 5

**Clarity, Quality, Novelty And Reproducibility:**

Clarity: The paper is well written and easy-to-follow.
Quality: The technical quality is good. The method proposed in this paper is empirically evaluated.
Novelty: The idea of this paper is somewhat plain.
Reproducibility: Model and experiment settings have been clarified to contribute to the reproducibility, but it would be better if the source code is publicly available.

**Strength And Weaknesses:**

Strength:
S1: The idea of augmenting federated learning from the perspective of CL periods is easy-to-implement but effective.
S2: The proposed FedCL framework is also simple and effective. It complements with state-of-the-art FL frameworks such as the popular FedAvg. Experimental results show FedCL improves the test accuracy with the same level of communication rounds.

Weaknesses:
W1: The novelty contribution of this paper could be limited. From the perspective of CL periods, the setting of FL has no intrinsic difference from the distributed machine learning, which has been studied in the cited references. Thus, although the idea of this paper is quite effective, it could be just a basic and straightforward solution.
W2: The proposed FGN metric is somewhat intuitive. While the result shown in Figure 16 shows that FGN is more lightweight than existing FedFIM metric, the test accuracy improvement of these two metrics is not reported.


**Summary Of The Paper:**

This paper presents a CL periods-aware federated learning framework to adaptively select the amount of clients.


**Summary Of The Review:**

The paper proposes a simple but effective framework to adaptively control the clients number in FL. The consideration of CL periods in FL is significant and the effectiveness of the proposed method is empirically evaluated. However, the idea of applying CL periods detection to distributed ML is a little plain, the proposed FGN metric is also intuitive. Consequently, I recommend this paper a borderline score.

---

> ### Author Response · Authors · 2022-11-13
> **Author Response to Reviewer 29oH**
>
> Thank you very much for your review and constructive comments. Here we would like to address the reviewer's concerns and hope that can help raise the rating of our paper. The detailed responses are as follows:
>
> **Your comment:** “W1: The novelty contribution of this paper could be limited. From the perspective of CL periods, the setting of FL has no intrinsic difference from the distributed machine learning, which has been studied in the cited references. Thus, although the idea of this paper is quite effective, it could be just a basic and straightforward solution. ”
>
> **Our response:** Thank you for your insightful comments. See our response to the “Common comment 1” above.
>
> **Your comment:** “W2: The proposed FGN metric is somewhat intuitive. While the result shown in Figure 16 shows that FGN is more lightweight than existing FedFIM metric, the test accuracy improvement of these two metrics is not reported.”
>
> **Our response:** Thank you for your insightful comments. See our response to the “Common comment 2” above.

---

> > ### Author Response · Authors · 2022-11-16
> > **Follow-up with Reviewer 29oH**
> >
> > Since the reviewer-author discussion period is ending soon, we just wanted to check in and ask if our rebuttal clarified and answered your questions. We would be very happy to engage further if there are additional questions.
> >
> > Also, we wanted to check if our additional clarifications regarding the merits of the paper would convince the reviewer to raise the score. Thank you!

---

### Author Response · Authors · 2022-11-13
**General Response (1/2)**

We thank all reviewers for the constructive comments and suggestions. We first answer two questions on (i) the novelty of our proposed critical learning (CL) periods aware federated learning (FL) framework; and (ii) the test accuracy using FedFIM. Following these, we are providing herewith a response to each of the reviewers explaining how we have addressed their comments. We have made specific changes based on the comments and suggestions of the reviewers. We have shown all changes along with additional experimental results in **blue color text** in the revised version. We believe that, upon this revision, the quality of our paper has significantly improved, and we hope this version and response could help raise the rating of our paper.

**Common Comment 1 (Reviewer 29oH, dEoL):** The novelty of the critical learning periods aware FL framework.

**Our response to Common Comment 1:** The connection between centralized/decentralized optimization and generalization of deep neural network training is not fully understood. A line of work on understanding the early phase of training has recently emerged as a promising avenue for studying this connection. For instance, Achille et al. 2019 is the among the first to highlight the presence of CL periods in deep neural network training in centralized settings, which is decisive for the final generalization ability. Gang et al. 2022 is the first to point out the existence of CL periods in FL settings, and empirically demonstrate that the rapid change in the local shape of the loss surface in the initial learning phases.

In this work, we reach a similar conclusion for FL: we identify the importance of the initial learning/training phases through the lens of a new metric called federated gradient norm (FGN). As discussed in the paper, FGN can identify the CL periods in FL training almost identical as those identified by FedFIM in Gang et al. 2022 (see Figure 1). However, our FGN is more computationally efficient than FedFIM (see Figure 16) and hence can be easily leveraged in the generalization of FL algorithm design, e.g., the client selection problem considered in this paper.

Though the idea seems quite intuitive, however, many questions remain open, including but not limited to how to efficiently identify CL periods, how to properly leverage them into improving the algorithm efficiency (E.g., client selection for model accuracy, communication efficiency, etc.), and how to theoretically analyze the generalization performance for deep/federated learning. In this work, we provide the first step to connect the generalization of FL algorithm design with the existence of CL periods. This differs significantly from most of existing algorithms in FL settings, which mainly focus on the federated optimization and ignore the impact of the underlying deep neural networks that are used to implement and train the designed FL algorithms.  This paper is a first step to close the gap between the observation of CL periods in FL and the goal of more efficient training and improved model accuracy. Our obtained insights shed light on improving FL of other machine learning models on other popular techniques such as gradient compression/quantization, personalization, attacks and defense, etc.

---

> ### Author Response · Authors · 2022-11-13
> **General Response (2/2)**
>
> **Common Comment 2 (Reviewer 29oH, 8m8J, phLC):** The test accuracy of FedCL using the metric of FedFIM.
>
> **Our response to Common Comment 2:** As mentioned in the paper as well as our response to Common Comment 1, our key idea to improve the model accuracy for efficient FL training to the connect the FL optimization with the underlying property in the deep neural networks that are used for FL training. To achieve this goal, the first step is how to efficiently identify the CL periods in the deep neural networks.
>
> To our best knowledge, Gang et al. 2022 provides the first comprehensive study on identifying CL periods in FL settings. Though Gang et al. 2022 reveals the existence of CL periods, their metric named FedFIM is computationally expensive and hence are hardly to be leveraged into the online algorithm (i.e., client selection) design. Therefore, our first contribution is to introduce the FGN metric that can equivalently identify the CL periods in FL as those identified by using the FedFIM in Gang et al. 2022. As shown in Figure 1, our FGN indeed is as useful as FedFIM, i.e., the CL periods identified by FGN are almost the same as those identified by FedFIM, but is more computationally efficient (as observed in Figure 16). Hence, we use FGN to identify CL periods in the design of FedCL, rather than using FedFIM. That is the reason that we did not report the test accuracy of FedCL when using FedFIM to identify CL periods.
>
> We thank the reviewers for this insightful comment. Since the CL periods identified by using FedFIM and FGN are almost the same, the test accuracy of FedCL should be almost the same.  We support this statement with additional experimental results. We denote FedCL(FedFIM) and FedCL(FGN) as the FedCL when leveraging the CL periods identified by using the FedFIM and FGN, respectively. As reported in the new Figure 17 in Appendix A.2.1, the test accuracy of FedCL(FedFIM) and FedCL(FGN) is almost identical, which is as expected.

---

### Decision · Program_Chairs · 2023-01-20

**Decision:**

Reject

**Justification For Why Not Higher Score:**

- Not much novelty, other than leveraging of known phenomenon of critical learning period in a federated setup.
- No explanation/motivation on why proposed FGN detects critical learning phase.
- Some discrepancy how communication cost is compared across different methods.

**Justification For Why Not Lower Score:**

N/A

**Metareview: Summary, Strengths And Weaknesses:**

This paper attempts to improve distributed training of deep networks. Towards this goal the authors leverage the known phenomenon of critical learning period in federated setup to achieve higher test accuracies across multiple datasets. In particular, a new metric called FGN is proposed to identify critical learning period and then adaptively select the amount of clients participating  based on it. We appreciate the extensive empirical studies carried out by the authors, but reviewers identified issues with the intricacies. We thank the authors and reviewers for actively engaging in discussion towards improving the paper, but still many of reviewer questions remain unanswered. It would be beneficial to understand more about the FGN metric and why FGN is a good metric for identifying critical learning period. Also highlighting key challenges about critical learning in a federated setup compared to centralized setting would make the paper stronger. Finally, it is not clear where the gains are coming from in the empirical studies. Is it due to the increased communication? The communication cost measured only in terms of rounds doesn't account for increased client participation in the proposed method. Carrying out the communication cost ablation as requested by the reviewers is crucial to demonstrate effectiveness of the proposed method.

**Summary Of Ac-Reviewer Meeting:**

N/A